# Parsimonious Demonstrations and Fine-Tuning for Large Language Models

## Abstract

Large language models (LLMs) have achieved impressive few-shot performance when provided with a small number of demonstrations as input context. In this paper, we systematically investigate what types of demonstrations are highly effective. Unlike prior approaches that select demonstrations based on similarity or diversity without considering LLMs, our insight is that the effectiveness of demonstrations depends on the specific LLMs used. In light of this, we introduce FEEDER (FEw yet Essential Data minER), a novel data miner that evaluates "sufficiency" and "necessity" of incorporating demonstrations as the context, taking into account the LLMs in use. The set of demonstrations that are both sufficient and necessary, referred to as parsimonious sets, can be viewed as a core subset of the training dataset, containing the most informative samples. Since evaluating all possible subsets is impractical, we devise novel tree-based search algorithms for identifying parsimonious sets. We demonstrate that these sets can serve two primary purposes. One is in-context learning, where FEEDER allows demonstration retrievers to operate on a subset rather than the entire training dataset, thus avoiding the retrieval of insufficient or unnecessary demonstrations. The other is fine-tuning, where fine-tuning LLMs on the set identified by FEEDER can yield improved performance while also reducing computational costs. Our empirical results on six text classification datasets and four LLM bases (ranging from 335M to 7B) consistently demonstrate: (i) In terms of few-shot inference, FEEDER allows the LLMs to achieve superior (or comparable) performance while utilizing only half the size of the input training data. (ii) With fine-tuning setting, FEEDER can significantly improve the LLM's performance.

## 1 Introduction

Large language models (LLMs), e.g., GPT-3 (Brown et al., 2020) and Llama (Touvron et al., 2023), have demonstrated impressive performance across a wide range of tasks by employing few-shot inference, often referred as in-context learning (Brown et al., 2020; Dong et al., 2022). This approach avoids the computational expense associated with fine-tuning LLMs. Here, the core challenge is how to select the most effective demonstrations from a large training set. Early methods (Qiu et al., 2022; Liu et al., 2021; Rubin et al., 2021; Wang et al., 2022) primarily selected demonstrations based on relevance, using similarity scores between each demonstration and the input question. Recent studies (Levy et al., 2022) have also incorporated diversity along with similarity, acknowledging that measuring each example in isolation can lead to sub-optimal results. We argue that all these previously used metrics should be thoroughly revised in the new era of LLMs because they measure each data instance regardless of the LLMs in use.

Our main idea is that the effectiveness of demonstrations should depend on the LLMs used. In light of this, we propose a data miner, named FEEDER (FEw yet Essential Data minER), to determine what types of demonstrations are effective and develop efficient methods for their selection.

We begin by investigating *sufficiency* and *necessity* of prompting each demonstration. Sufficiency assesses whether prompting a demonstration enhances LLM performance on domain-specific tasks, while necessity gauges whether a newly considered demonstration provides redundant information compared to those already included. The resulting sets of selected demonstrations, deemed sufficient and necessary, form what we term *parsimonious sets*.

To efficiently select a parsimonious set from the training dataset, the exhaustive enumeration and evaluation of all possible subsets is impractical. Instead, we devise tree-based algorithms to first examine whether each demonstration is sufficient and necessary to represent other demonstrations, and then form a parsimonious set. Our approach functionality can be regarded as an unsupervised core-set selection method, producing a subset of training instances that are highly informative for the downstream tasks including in-context learning and fine-tuning. Under the in-context learning setting, FEEDER can collaborate with various demonstration retrievers, employing the parsimonious set as the retrieval pool instead of the entire training dataset to generate n-shot demonstrations. Besides, we also demonstrate that the parsimonious set can enhance the fine-tuning process. Typically, we evaluate the LLM performance in terms of using one epoch (called warm-up), since more extended epochs usually require a larger amount of multiple-domain data and greater computational resources. The above observations together can lead to a novel bi-level framework, in which we formulate the parsimonious set selection and the LLM fine-tuning as a unified bi-level optimization problem. It comprises an outer level for extracting the parsimonious set using a frozen LLM and an inner level for fine-tuning the LLM with fixed plug-in data. This process could be iterated with the tuned LLM serves for the new parsimonious set selection in the next iteration.

While practical, we could pre-compute and store parsimonious sets to reduce computational overhead for downstream tasks. Considering that the training dataset can continuously expand, we develop an incremental update version for FEEDER. This algorithm allows FEEDER to run selectively to the newly added or modified examples, avoiding the need to recompute every example. Our empirical results, which span across six text classification datasets, four LLM bases (ranging from 335M to 7B), and three popular demonstration retrievers (including random, similarity-based, and diversity-based retrievers), demonstrate the effectiveness and efficiency of FEEDER. It efficiently selects a parsimonious set from the entire training dataset, saving nearly half of the data size. Furthermore, utilizing this selected subset, instead of the full training dataset, consistently yields better or comparable performance in 1-shot, 2-shot, 5-shot, and 10-shot settings. Our results also show that fine-tuning LLMs on the parsimonious set consistently leads to significant improvements compared to fine-tuning on the entire training dataset. We further expand the evaluation of FEEDER to reasoning and semantic-parsing tasks using GPT-6B providing consistent results with the trends observed in the text classification task.

## 2    A DATA-CENTRIC VIEW FOR IN-CONTEXT LEARNING AND FINE-TUNING

We begin by describing two different contexts where FEEDER operates: the in-context learning setting and the fine-tuning setting. In this paper, we explore both of them from a data-centric viewpoint (Strickland, 2022), where *data quality* outweighs *data quantity*.

In the in-context learning setting, we are given a training dataset $\mathcal{D}_{\texttt{TRAIN}} = \{(\boldsymbol{x}_n, \boldsymbol{y}_n)\}_{n=1}^{N}$ consisting of pairs of input data (e.g., questions) and output labels (i.e., answers). Additionally, we have a test dataset $\mathcal{D}_{\texttt{TEST}} = \{(\boldsymbol{x}_m, \boldsymbol{y}_m)\}_{m=1}^{M}$, where we assume that $\mathcal{D}_{\texttt{TRAIN}}$ share the same support set (Yosida, 2012) with $\mathcal{D}_{\texttt{TEST}}$. Our goal is to develop a demonstration miner that extracts a subset of training examples, denoted as $\widetilde{\mathcal{D}}_{\texttt{TRAIN}} \subset \mathcal{D}_{\texttt{TRAIN}}$. We use $\Psi_{\texttt{LLM}} : \mathbb{X} \times \mathbb{D} \to \mathbb{Y}$ to represent a LLM using selected demonstrations as the context. Here, $\boldsymbol{x}. \in \mathbb{X}$ is an input text, $\boldsymbol{y}. \in \mathbb{Y}$ is the corresponding output, and $(\boldsymbol{x}., \boldsymbol{y}.) \in \mathbb{D}$ is one demonstration. Formally, our objective is to minimize:

$$\mathcal{L}\big(\widetilde{\mathcal{D}}_{\texttt{TRAIN}}, \mathcal{D}_{\texttt{TEST}}; \Psi_{\texttt{LLM}}^{*}\big) = \sum_{(\boldsymbol{x}_m, \boldsymbol{y}_m) \in \mathcal{D}_{\texttt{TEST}}} \ell\Big(\Psi_{\texttt{LLM}}^{*}(\boldsymbol{x}_m, \widetilde{\mathcal{D}}_{\texttt{TRAIN}}), \boldsymbol{y}_m\Big), \tag{1}$$

where $\ell(\cdot)$ is the given task-specific loss function, and $\Psi_{\texttt{LLM}}^{*}(\cdot)$ means that the LLM is frozen. However, we do not have direct access to the test dataset $\mathcal{D}_{\texttt{TEST}}$, making it impractical to optimize the demonstration selection directly by minimizing $\mathcal{L}(\widetilde{\mathcal{D}}_{\texttt{TRAIN}}, \mathcal{D}_{\texttt{TEST}}; \Psi_{\texttt{LLM}}^{*})$.

Instead, we re-consider the demonstration miner task to select a set of high-quality demonstrations from the training samples. Our key idea is that high-quality training examples $\widetilde{\mathcal{D}}_{\texttt{TRAIN}}$ should be both representative of the entire training dataset $\mathcal{D}_{\texttt{TRAIN}}$ and as minimal in size as possible. Formally, we formulate this objective as:

$$\min_{\widetilde{\mathcal{D}}_{\texttt{TRAIN}} \subset \mathcal{D}_{\texttt{TRAIN}}} |\widetilde{\mathcal{D}}_{\texttt{TRAIN}}|, \text{ s.t. } \mathcal{L}(\widetilde{\mathcal{D}}_{\texttt{TRAIN}}, \mathcal{D}_{\texttt{TRAIN}}; \Psi_{\texttt{LLM}}^{*}) \leq \mathcal{L}(\mathcal{D}_{\texttt{TRAIN}}, \mathcal{D}_{\texttt{TRAIN}}; \Psi_{\texttt{LLM}}^{*}). \tag{2}$$

This formulation ensures that $\widetilde{\mathcal{D}}_{\texttt{TRAIN}}$ is not only sufficient but also necessary to represent $\mathcal{D}_{\texttt{TRAIN}}$, thus removing redundant data points to save computation costs meanwhile maintaining LLM performance.

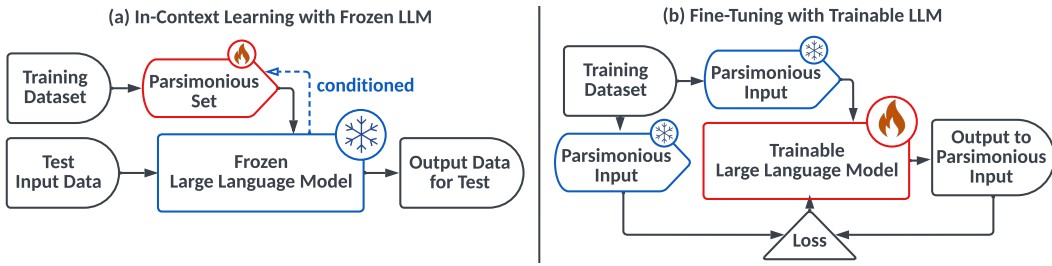

Figure 1: Overview of FEEDER that operates effectively within both in-context learning and fine-tuning contexts. In the in-context learning setting, depicted in (a), FEEDER selects a parsimonious set from the training dataset, and this selected set is characterized by its sufficiency and necessity conditioned on the frozen LLM. In the fine-tuning setting, shown in (b), FEEDER allows the LLM to be tuned on the fixed parsimonious set, and this parsimonious set is intentionally selected to be a faithful representation of the training dataset, with the dual objectives of maintaining data quality and minimizing computational expenses.

The above data-centric viewpoint of mining a subset of high-quality demonstrations also can be applied to fine-tune LLMs. Given a training dataset $\mathcal{D}_{\texttt{TRAIN}}$ and a test dataset $\mathcal{D}_{\texttt{TEST}}$, our objective is to minimize $\mathcal{L}(\emptyset, \widetilde{\mathcal{D}}_{\texttt{TRAIN}}; \Psi_{\texttt{LLM}})$, where $\widetilde{\mathcal{D}}_{\texttt{TRAIN}}$ is expected to be both sufficient and necessary to represent $\mathcal{D}_{\texttt{TRAIN}}$, which allows us to fine-tune the LLM effectively while reducing computation costs. In this case, the LLM $\Psi_{\texttt{LLM}}$ is usually trainable, and our goal can be formulated as:

$$\min_{\Psi_{\texttt{LLM}}} \mathbb{E}_{(\boldsymbol{x}_n, \boldsymbol{y}_n) \in \widetilde{\mathcal{D}}_{\texttt{TRAIN}}^*} [\ell\Big(\Psi_{\texttt{LLM}}(\boldsymbol{x}_n, \emptyset), \boldsymbol{y}_n\Big)], \qquad (3)$$

where $\widetilde{\mathcal{D}}_{\texttt{TRAIN}}^*$ means that the selected parsimonious set is fixed during fine-tuning.

**Bridging Parsimonious Set Selection and LLM Fine-Tuning into a Unified Bi-level Optimization Framework.** On the outer level, following Eq. (2), we optimize the parsimonious set $\widetilde{\mathcal{D}}_{\texttt{TRAIN}}$ in the context of a frozen LLM $\Psi_{\texttt{LLM}}^*$ (as illustrated in Figure 1(a)), while on the inner level, following Eq. (3), we optimize the LLM $\Psi_{\texttt{LLM}}$ using the fixed parsimonious set $\widetilde{\mathcal{D}}_{\texttt{TRAIN}}^*$ (as shown in Figure 1(b)). It is noteworthy that this bi-level optimization procedure is amenable to repetition, allowing for iterative refinement of both the selected parsimonious set and the tuned LLM.

**Connections to Existing Work.** With the growing capabilities of LLMs, a new paradigm, called in-context learning (Dong et al., 2022), has emerged, where the core challenge is selecting appropriate training examples as the context, often referred to as "demonstrations". Previous solutions have revolved around constructing either parameter-free retrieval mechanisms (Wang et al., 2022; Zemlyanskiy et al., 2022) or neural-based retrieval methods (Pasupat et al., 2021; Liu et al., 2021; Gupta et al., 2021; Rubin et al., 2021). However, as pointed out in (Levy et al., 2022), greedily retrieving demonstrations solely based on their similarity to the input question can result in sub-optimal performance. Different from the above retrievers, which do not consider the specific LLM in use, FEEDER evaluates the sufficiency and necessity of each demonstration's incorporation based on the LLM used. Our bi-level optimization framework is also related to core-set selection techniques in active learning (Feldman, 2020; Guo et al., 2022). Prior literature (Dor et al., 2020) has summarized existing state-of-the-art strategies for models like BERT (Devlin et al., 2018), including uncertainty sampling (based on entropy metric) (Lewis, 1995; Gal & Ghahramani, 2016) and diversity sampling (focused on diversity metric) (Gissin & Shalev-Shwartz, 2019). In contrast, FEEDER identifies the parsimonious set, which can serve a dual purpose: it can be used as candidate input contexts and, alternatively, to fine-tune the LLM.

We provide a comprehensive summary of other existing methods in Appendix A.

## 3 SUFFICIENCY AND NECESSITY: FROM INSTANCE TO SET

Let $X, C$ denote variables for the input and the context (i.e., selected demonstrations). We introduce $Y$, a boolean variable, to represent whether the output to the input is correct. For simplicity, we use $Y_{\boldsymbol{x}_n} = 1$ to denote $Y = 1 | X = \boldsymbol{x}_n$, meaning that the LLM generates the correct output for the input $\boldsymbol{x}_n$. Similarly, $Y_{\boldsymbol{x}_n} = 0$, equivalent to $Y = 0 | X = \boldsymbol{x}_n$, indicates that LLM produces an incorrect output for $\boldsymbol{x}_n$. For convenience, we introduce $S$, a variable to record the original status of the LLM before new plug-in and unplug operations (denoted as $\texttt{plug}(\cdot)$ and $\texttt{unplug}(\cdot)$ respectively).

We begin by considering the relationship between two examples, denoted as $(\boldsymbol{x}_n, \boldsymbol{y}_n)$ and $(\boldsymbol{x}_m, \boldsymbol{y}_m)$.

*Sufficiency* relationship is introduced to assess whether plugging in one data point is adequate for the LLM to produce the correct answer to another data point. Formally, this relationship is expressed as:

**Definition 1 (Sufficiency).** *Given tuple* $(X, Y, C, S)$, *a training example* $(\boldsymbol{x}_n, \boldsymbol{y}_n)$ *is considered a sufficient instance for another example* $(\boldsymbol{x}_m, \boldsymbol{y}_m)$, *if the following equation holds:*

$$Y_{\boldsymbol{x}_m} = 1 | \texttt{plug}((\boldsymbol{x}_n, \boldsymbol{y}_n)); C = \emptyset, S, \tag{4}$$

*where $S$ can be any value. It means that prior to plugging in $(\boldsymbol{x}_n, \boldsymbol{y}_n)$, there is no existing plugged-in data as contexts (i.e., $C = \emptyset$), and when we do plug-in $(\boldsymbol{x}_n, \boldsymbol{y}_n)$ (i.e., $\texttt{plug}((\boldsymbol{x}_n, \boldsymbol{y}_n))$), it results in the LLM providing the correct output (i.e., $Y_{\boldsymbol{x}_m} = 1$).*

*Necessity* relationship is introduced to assess whether it is necessary to retain a particular plugged-in data point to maintain the correct output of another data point. Its formal definition can be written as:

**Definition 2 (Necessity).** *Given tuple* $(X, Y, C, S)$, *a training example* $(\boldsymbol{x}_n, \boldsymbol{y}_n)$ *is considered a necessary instance for another example* $(\boldsymbol{x}_m, \boldsymbol{y}_m)$, *if the following equation holds:*

$$Y_{\boldsymbol{x}_m} = 0 | \texttt{unplug}((\boldsymbol{x}_n, \boldsymbol{y}_n)); C = ((\boldsymbol{x}_n, \boldsymbol{y}_n)), S = (Y_{\boldsymbol{x}_m} = 1). \tag{5}$$

*It means that prior to unplugging $(\boldsymbol{x}_n, \boldsymbol{y}_n)$, there is plugged-in data as contexts (i.e., $C = ((\boldsymbol{x}_n, \boldsymbol{y}_n))$), and the LLM's output to $\boldsymbol{x}_m$ is correct (i.e., $S = (Y_{\boldsymbol{x}_m} = 1)$). However, when we do unplug $(\boldsymbol{x}_n, \boldsymbol{y}_n)$ (i.e., $\texttt{unplug}((\boldsymbol{x}_n, \boldsymbol{y}_n))$), it causes the LLM offering an incorrect output (i.e., $Y_{\boldsymbol{x}_m} = 0$).*

The above definitions of sufficiency and necessity metrics, operating on the instance level, are further clarified with examples in Appendix B.1. Extending these definitions to the set level, a sufficient set signifies that plugging in a specific set is adequate to ensure the correct outputs for all examples in another set, while a necessary set implies that removing any example from this set would result in incorrect answers for at least one example within another set. Formal definitions for the above set-level metrics, along with examples, are available in Appendix B.2.

Taking into account both the sufficiency and necessity metrics, we define a subset of the training dataset $\mathcal{D}_{\texttt{TRAIN}}$, which is both sufficient and necessary to effectively represent the entire training dataset $\mathcal{D}_{\texttt{TRAIN}}$, as parsimonious set $\mathcal{D}_{\texttt{FEED}}$. Formally, we define $\mathcal{D}_{\texttt{FEED}}$ as follows:

**Definition 3 (Parsimonious Set).** *Given tuple* $(X, Y, C, S)$ *and the training dataset* $\mathcal{D}_{\texttt{TRAIN}}$, *a subset of* $\mathcal{D}_{\texttt{TRAIN}}$, *is considered as a parsimonious set, if the following conditions are satisfied:*

*(i)* $Y_{(\boldsymbol{x}_1 \dots, \boldsymbol{x}_N)} = \mathbf{1}_N | \texttt{plug}(\mathcal{D}_{\texttt{FEED}}); C = \emptyset, S$ *holds, where $S$ can be any value. $\mathbf{1}_N$ and $\mathbf{0}_N$ denotes $N$-dimensional vectors whose elements are all 1s and 0s. It implies that plugging in the parsimonious set $\mathcal{D}_{\texttt{FEED}}$ alone is sufficient to maintain $Y_{(\boldsymbol{x}_1 \dots, \boldsymbol{x}_N)} = \mathbf{1}_N$.*

*(ii)* $Y_{(\boldsymbol{x}_1 \dots, \boldsymbol{x}_N)} \neq \mathbf{1}_N | \texttt{unplug}(\mathcal{D}'_{\texttt{FEED}} \cup \overline{\mathcal{D}}_{\texttt{FEED}}); C = \mathcal{D}_{\texttt{TRAIN}}, S = (Y_{(\boldsymbol{x}_1 \dots, \boldsymbol{x}_N)} = \mathbf{1}_N)$ *holds for any subset of $\mathcal{D}_{\texttt{FEED}}$ (denoted as $\mathcal{D}'_{\texttt{FEED}}$) and $\overline{\mathcal{D}}_{\texttt{FEED}} = \mathcal{D}_{\texttt{TRAIN}} - \mathcal{D}_{\texttt{FEED}}$. It indicates that given $\overline{\mathcal{D}}_{\texttt{FEED}}$ is unplugged, then unplugging any example in $\mathcal{D}_{\texttt{FEED}}$ would make the plugged-in data not sufficient to keep the outputs correct for all the inputs. Namely, plugging in $\mathcal{D}_{\texttt{FEED}}$ is necessary for maintaining $Y_{(\boldsymbol{x}_1 \dots, \boldsymbol{x}_N)} = \mathbf{1}_N$.*

We illustrate the concept of the parsimonious set through specific examples and establish the relationships between the parsimonious set and the concepts of sufficient and necessary sets in Appendix B.2.

## 4 SEARCHING PARSIMONIOUS SET: FROM CONCEPT TO PRACTICE

Directly searching $\mathcal{D}_{\texttt{FEED}}$ using an exhaustive approach, which involves evaluating all possible subsets of $\mathcal{D}_{\texttt{TRAIN}}$, is impractical due to its computational complexity, requiring $O(2^N)$ computations.

To streamline the computation, we develop tree-search algorithms to extract sufficient and necessary sets separately from the given dataset. For convenience, we use $\mathcal{D}_{\texttt{IN}} = \{(\boldsymbol{x}_n, \boldsymbol{y}_n)\}_{n=1}^{N_{\texttt{IN}}}$ to denote the input set for these algorithms, and use $\mathcal{D}_{\texttt{OUT}}$ to denote the output set. The trees in our approach expand from the bottom to the top. We use the variable $K$ to represent the depth of these trees, which corresponds to the number of iterations. To be more specific, we use $k = 1, 2, \dots, K$ to refer to each $k$-th iteration, and during each $k$-th iteration, we generate the $(k+1)$-th layer of the tree.

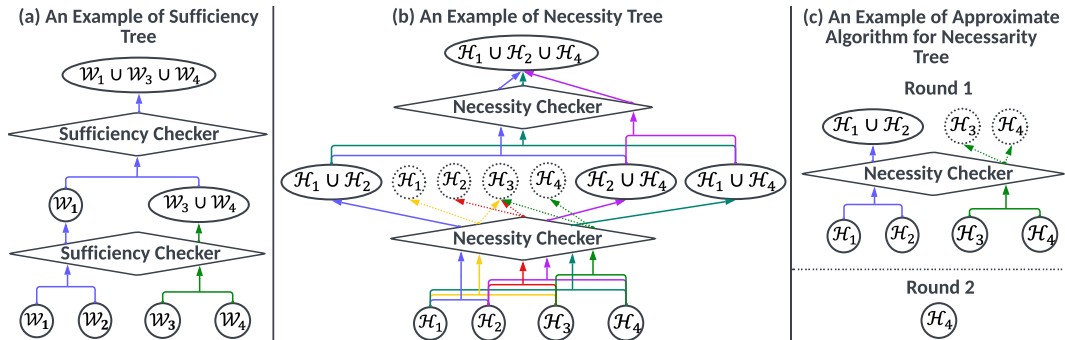

Figure 2: An illustrated example of mining a sufficient set and a necessary set through building sufficiency and necessity trees. In (a), we use a sufficiency checker to check each pair of nodes, and after checking, we remove them from $\mathscr{W}$; in other words, each node would only be checked once. In (b), we run a necessity checker to check each pair of nodes, and we do not remove them from $\mathscr{H}$; instead, we assign MAINTAIN signals to newly generated nodes and the node with the maximum size, and those nodes without MAINTAIN signals, circled with dashed lines, would be removed from $\mathscr{H}$. In (c), we approximate (b) by removing nodes after checking (similar to (a)), and we repeat the above process for multiple rounds, at the beginning of each round, we unplug all the previously selected data points. The repeat should stop until there is no or only one node in $\mathscr{H}_0$ (i.e., $\mathcal{H}_4$), and therefore, the result in (c) is $\mathcal{H}_1 \cup \mathcal{H}_2 \cup \mathcal{H}_4$, same as the result in (b).

**Searching Sufficient Set.** We begin by introducing the following assumption: sufficiency follows a transitive relationship among sets. Namely, if $\mathcal{D}_{\mathtt{A}}$ is a sufficient set for $\mathcal{D}_{\mathtt{B}}$, and $\mathcal{D}_{\mathtt{B}}$ is a sufficient set for $\mathcal{D}_{\mathtt{C}}$, then $\mathcal{D}_{\mathtt{A}}$ is considered a sufficient set for $\mathcal{D}_{\mathtt{C}}$. This assumption is based on the capability of LLMs to infer with a chain of thoughts (Wei et al., 2022). We also include case studies in Appendix E to illustrate the feasibility of the above assumptions.

We leverage the transitivity of sufficiency to build a sufficiency tree, where each node is a set of examples. Formally, we denote $\mathscr{W}_k$ as the set of nodes after the $k$-th iteration. We initialize $\mathscr{W}_0$ by assigning all the candidate examples in $\mathcal{D}_{\mathtt{IN}}$ as the bottom nodes:

$$\mathscr{W}_0 := \{\mathcal{W}_n := \{(\boldsymbol{x}_n, \boldsymbol{y}_n)\} | (\boldsymbol{x}_n, \boldsymbol{y}_n) \in \mathcal{D}_{\mathtt{IN}}\}. \tag{6}$$

During each $k$-th iteration, we employ a sufficiency checker to generate $\mathscr{W}_k$. This is accomplished by examining the sufficiency relationship between every pair of nodes, denoted as $\mathcal{W}_i$ and $\mathcal{W}_j$ from $\mathscr{W}_{k-1}$. In this evaluation, we assess whether the following equation holds true by assigning $\mathcal{W}_i$ and $\mathcal{W}_j$ as $\mathcal{W}_{\mathtt{IN}}$ and $\mathcal{W}_{\mathtt{OUT}}$, or vice versa.

$$Y_{(\{\boldsymbol{x}_n | \boldsymbol{x}_n \in \mathcal{W}_{\mathtt{OUT}}\})} = \mathbf{1}_{|\mathcal{W}_{\mathtt{OUT}}|} | \mathtt{plug}(\mathcal{W}_{\mathtt{IN}}); C = \emptyset, S, \tag{7}$$

where $S$ can be any value. If Eq. (7) holds, it signifies that plugging in $\mathcal{W}_{\mathtt{IN}}$ is sufficient for the LLM to generate the correct output to any input in $\mathcal{W}_{\mathtt{OUT}}$. There are three possible scenarios: (i) Reciprocal Sufficiency: If both $\mathcal{W}_i$ and $\mathcal{W}_j$ are sufficient set for each other, then we select the one with fewer elements to append to $\mathscr{W}_k$. (ii) One-sided Sufficiency: If only one of $\mathcal{W}_i$ and $\mathcal{W}_j$ is a sufficient set for the other, then we append the sufficient set to $\mathscr{W}_k$. (iii) Insufficiency: If neither $\mathcal{W}_i$ nor $\mathcal{W}_j$ is a sufficient set, we append $\mathcal{W}_i \cup \mathcal{W}_j$ to $\mathscr{W}_k$. After performing the above calculations for each pair of nodes, we remove them from $\mathscr{W}_{k-1}$. When there is only one element left in $\mathscr{W}_{k-1}$, it is directly appended to $\mathscr{W}_k$. This process continues until $\mathscr{W}$ contains only one element, which is denoted as $\mathcal{W}_{\mathtt{SUFFICIENT}} \in \mathscr{W}_K$. We then assign $\mathcal{D}_{\mathtt{OUT}}$ as $\mathcal{D}_{\mathtt{OUT}} = \mathcal{W}_{\mathtt{SUFFICIENT}}$.

We illustrate the above process with an example in Figure 2(a), and running the sufficiency tree search algorithm requires $O(\log_2^{|\mathcal{D}_{\mathtt{IN}}|})$ iterations.

**Searching Necessary Set.** Notice that if unplugging set $\mathcal{D}_{\mathtt{A}}$ causes the output to at least one input in set $\mathcal{D}_{\mathtt{C}}$ to change from correct to wrong, then unplugging the union of sets $\mathcal{D}_{\mathtt{A}}$ and $\mathcal{D}_{\mathtt{B}}$ also can not maintain the correctness of all outputs in set $\mathcal{D}_{\mathtt{C}}$. Our goal here is to extract the unnecessary part from the input set $\mathcal{D}_{\mathtt{IN}}$ so that the remaining part becomes a necessary set. For this purpose, we create a necessity tree, where each node represents a set of unnecessary examples. We use $\mathscr{H}_k$ to denote a set of nodes after the $k$-th iteration. We initialize $\mathscr{H}_0$ by identifying all examples in $\mathcal{D}_{\mathtt{IN}}$ for which unplugging them individually does not affect the LLM's performance. Formally, we construct $\mathscr{H}_0$ as follows. $\mathscr{H}_0 := \{\mathcal{H}_n := \{(\boldsymbol{x}_n, \boldsymbol{y}_n)\}\}$ where $(\boldsymbol{x}_n, \boldsymbol{y}_n) \in \mathcal{D}_{\mathtt{IN}}$ satisfies:

$$Y_{(\{\boldsymbol{x}_{n'} | \boldsymbol{x}_{n'} \in \mathcal{D}_{\mathtt{IN}}\})} = \mathbf{1}_{|\mathcal{D}_{\mathtt{IN}}|} | \mathtt{unplug}((\boldsymbol{x}_n, \boldsymbol{y}_n)); C = \mathcal{D}_{\mathtt{IN}}, S = (Y_{(\{\boldsymbol{x}_{n'} | \boldsymbol{x}_{n'} \in \mathcal{D}_{\mathtt{IN}}\})} = \mathbf{1}_{|\mathcal{D}_{\mathtt{IN}}|}). \tag{8}$$

During each $k$-th iteration, we employ a necessity checker to generate $\mathscr{H}_k$ by examining the necessity relationship between each pair of nodes (denoted as $\mathcal{H}_i$ and $\mathcal{H}_j$ in $\mathscr{H}_{k-1}$). Here, we verify whether

Table 1: Performance comparisons on text classification datasets for the in-context learning setting. We report both the mean and variance of accuracy using four different seeds and four different permutations of n-shots. See Table 4 for more extended results on datasets COLA, SST-2, and FPB.

| $\Psi_{\text{LLM}}(\cdot)$ | $\widetilde{\mathcal{D}}_{\text{TRAIN}}$ | $n$ | SUBJ | | | SST-5 | | | TREC | | |
|---|---|---|---|---|---|---|---|---|---|---|---|
| | | | RAN | SIM | DIV | RAN | SIM | DIV | RAN | SIM | DIV |
| MED | $\mathcal{D}_{\text{TRAIN}}$ | 1 | 41.3 $_{(7.2)}$ | 41.1 $_{(0.1)}$ | 41.1 $_{(0.1)}$ | 14.5 $_{(6.1)}$ | 22.7 $_{(0.2)}$ | 22.7 $_{(0.2)}$ | 19.4 $_{(6.4)}$ | 42.8 $_{(0.1)}$ | 42.8 $_{(0.1)}$ |
| | | 2 | 47.3 $_{(7.2)}$ | 62.8 $_{(0.1)}$ | 71.9 $_{(0.2)}$ | 18.0 $_{(5.8)}$ | 25.6 $_{(0.1)}$ | 23.7 $_{(0.2)}$ | 21.4 $_{(4.7)}$ | 57.2 $_{(0.2)}$ | 51.4 $_{(0.1)}$ |
| | | 5 | 51.8 $_{(5.5)}$ | 85.8 $_{(0.3)}$ | 70.1 $_{(0.2)}$ | 26.5 $_{(5.3)}$ | 32.3 $_{(0.2)}$ | 27.8 $_{(0.1)}$ | 37.6 $_{(5.1)}$ | 66.0 $_{(0.3)}$ | 61.4 $_{(0.3)}$ |
| | | 10 | 62.4 $_{(5.0)}$ | 88.0 $_{(0.2)}$ | 78.2 $_{(0.1)}$ | 14.9 $_{(3.9)}$ | 35.3 $_{(0.1)}$ | 30.4 $_{(0.2)}$ | 53.0 $_{(5.2)}$ | 71.4 $_{(0.2)}$ | 65.8 $_{(0.3)}$ |
| | $\mathcal{D}_{\text{FEED}}$ | 1 | **42.8** $_{(2.4)}$ | **44.9** $_{(1.1)}$ | **44.9** $_{(1.1)}$ | 15.4 $_{(5.2)}$ | 23.7 $_{(1.7)}$ | 23.7 $_{(1.7)}$ | **37.4** $_{(3.6)}$ | **48.4** $_{(1.6)}$ | **48.4** $_{(1.6)}$ |
| | | 2 | **55.9** $_{(3.3)}$ | 63.4 $_{(1.6)}$ | **74.7** $_{(0.9)}$ | **20.9** $_{(4.7)}$ | **27.9** $_{(1.1)}$ | **25.8** $_{(1.3)}$ | 27.6 $_{(3.2)}$ | 56.8 $_{(2.2)}$ | 52.2 $_{(1.9)}$ |
| | | 5 | 57.5 $_{(4.0)}$ | **86.9** $_{(0.7)}$ | 69.8 $_{(1.0)}$ | **28.6** $_{(3.4)}$ | 33.2 $_{(1.8)}$ | 27.4 $_{(1.7)}$ | 44.6 $_{(3.0)}$ | 67.4 $_{(1.2)}$ | 61.8 $_{(1.3)}$ |
| | | 10 | 63.5 $_{(4.4)}$ | 88.1 $_{(1.3)}$ | **79.7** $_{(2.0)}$ | 17.6 $_{(2.2)}$ | **36.9** $_{(1.9)}$ | 29.8 $_{(1.7)}$ | 44.6 $_{(2.8)}$ | **74.6** $_{(1.4)}$ | 64.6 $_{(1.9)}$ |
| LAR | $\mathcal{D}_{\text{TRAIN}}$ | 1 | 42.5 $_{(5.2)}$ | 43.6 $_{(0.1)}$ | 43.6 $_{(0.1)}$ | 14.2 $_{(4.9)}$ | 25.2 $_{(0.1)}$ | 25.2 $_{(0.1)}$ | 21.0 $_{(4.6)}$ | 53.2 $_{(0.2)}$ | 53.2 $_{(0.2)}$ |
| | | 2 | 58.1 $_{(6.3)}$ | 88.3 $_{(0.2)}$ | 87.0 $_{(0.3)}$ | 18.1 $_{(5.1)}$ | 29.7 $_{(0.1)}$ | 24.4 $_{(0.2)}$ | 28.2 $_{(4.4)}$ | 62.6 $_{(0.2)}$ | 60.6 $_{(0.2)}$ |
| | | 5 | 66.7 $_{(4.5)}$ | 86.2 $_{(0.2)}$ | 86.7 $_{(0.1)}$ | 25.6 $_{(4.8)}$ | 34.1 $_{(0.1)}$ | 30.8 $_{(0.1)}$ | 35.4 $_{(5.7)}$ | 63.4 $_{(0.1)}$ | 64.6 $_{(0.1)}$ |
| | | 10 | 48.6 $_{(6.0)}$ | 85.9 $_{(0.1)}$ | 73.9 $_{(0.2)}$ | 28.7 $_{(4.2)}$ | 38.7 $_{(0.1)}$ | 36.6 $_{(0.1)}$ | 43.2 $_{(4.8)}$ | 66.0 $_{(0.1)}$ | 68.8 $_{(0.1)}$ |
| | $\mathcal{D}_{\text{FEED}}$ | 1 | 42.8 $_{(5.4)}$ | **46.4** $_{(0.4)}$ | **46.4** $_{(0.4)}$ | **18.7** $_{(3.0)}$ | 25.5 $_{(2.2)}$ | 25.5 $_{(2.2)}$ | 17.4 $_{(3.8)}$ | 52.6 $_{(2.1)}$ | 52.6 $_{(2.1)}$ |
| | | 2 | **63.1** $_{(4.5)}$ | 88.5 $_{(1.5)}$ | 86.8 $_{(1.3)}$ | **25.2** $_{(3.8)}$ | 29.7 $_{(1.9)}$ | 24.1 $_{(2.1)}$ | **34.6** $_{(3.5)}$ | 62.2 $_{(1.8)}$ | 59.4 $_{(2.0)}$ |
| | | 5 | **73.4** $_{(4.3)}$ | 86.2 $_{(1.9)}$ | 86.8 $_{(1.7)}$ | **39.3** $_{(2.9)}$ | **35.2** $_{(1.1)}$ | 31.0 $_{(1.2)}$ | **45.4** $_{(3.3)}$ | **65.5** $_{(1.5)}$ | 63.7 $_{(1.7)}$ |
| | | 10 | **52.0** $_{(3.8)}$ | 85.4 $_{(1.3)}$ | **75.6** $_{(1.2)}$ | **39.6** $_{(3.0)}$ | 38.8 $_{(1.8)}$ | **37.8** $_{(1.6)}$ | **55.8** $_{(3.8)}$ | **70.4** $_{(2.0)}$ | 68.6 $_{(1.7)}$ |
| NEO | $\mathcal{D}_{\text{TRAIN}}$ | 1 | 42.8 $_{(3.9)}$ | 42.1 $_{(0.1)}$ | 42.1 $_{(0.1)}$ | 12.8 $_{(2.7)}$ | 20.2 $_{(0.1)}$ | 20.2 $_{(0.1)}$ | 11.0 $_{(3.2)}$ | 57.2 $_{(0.2)}$ | 57.2 $_{(0.2)}$ |
| | | 2 | 48.5 $_{(4.2)}$ | 68.3 $_{(0.2)}$ | 72.6 $_{(0.3)}$ | 17.9 $_{(3.6)}$ | 26.9 $_{(0.1)}$ | 22.7 $_{(0.1)}$ | 17.6 $_{(3.1)}$ | 52.6 $_{(0.2)}$ | 42.2 $_{(0.2)}$ |
| | | 5 | 51.6 $_{(5.0)}$ | 80.5 $_{(0.2)}$ | 61.7 $_{(0.2)}$ | 19.0 $_{(3.9)}$ | 29.2 $_{(0.1)}$ | 25.1 $_{(0.1)}$ | 25.2 $_{(3.8)}$ | 66.4 $_{(0.1)}$ | 61.8 $_{(0.1)}$ |
| | | 10 | 48.5 $_{(5.8)}$ | 85.9 $_{(0.3)}$ | 81.9 $_{(0.1)}$ | 12.7 $_{(2.8)}$ | 33.7 $_{(0.2)}$ | 31.9 $_{(0.1)}$ | 41.6 $_{(4.4)}$ | 70.6 $_{(0.1)}$ | 69.0 $_{(0.1)}$ |
| | $\mathcal{D}_{\text{FEED}}$ | 1 | 43.2 $_{(4.0)}$ | **46.3** $_{(1.0)}$ | **46.3** $_{(1.0)}$ | **18.5** $_{(2.1)}$ | 20.6 $_{(1.8)}$ | 20.6 $_{(1.4)}$ | **18.2** $_{(2.4)}$ | 56.4 $_{(1.3)}$ | 56.4 $_{(1.3)}$ |
| | | 2 | **62.6** $_{(3.5)}$ | 68.4 $_{(1.5)}$ | **73.8** $_{(2.1)}$ | **19.7** $_{(2.7)}$ | 27.4 $_{(2.1)}$ | 22.8 $_{(1.8)}$ | **27.8** $_{(2.7)}$ | 54.0 $_{(1.4)}$ | **44.5** $_{(1.6)}$ |
| | | 5 | **69.4** $_{(5.6)}$ | 79.2 $_{(1.8)}$ | 61.9 $_{(1.3)}$ | 19.2 $_{(3.2)}$ | 30.2 $_{(2.7)}$ | **26.4** $_{(2.4)}$ | **50.4** $_{(3.2)}$ | 66.0 $_{(1.4)}$ | 61.6 $_{(1.5)}$ |
| | | 10 | **78.7** $_{(3.3)}$ | **87.2** $_{(1.7)}$ | 82.3 $_{(2.8)}$ | 15.4 $_{(2.4)}$ | 37.0 $_{(1.5)}$ | **34.5** $_{(1.9)}$ | 45.2 $_{(2.9)}$ | 72.8 $_{(1.4)}$ | 68.8 $_{(1.3)}$ |
| LLA | $\mathcal{D}_{\text{TRAIN}}$ | 1 | 42.9 $_{(6.6)}$ | 48.5 $_{(0.1)}$ | 48.5 $_{(0.1)}$ | 28.6 $_{(2.9)}$ | 29.7 $_{(0.1)}$ | 29.7 $_{(0.1)}$ | 35.2 $_{(3.7)}$ | 54.2 $_{(0.1)}$ | 54.2 $_{(0.1)}$ |
| | | 2 | 51.9 $_{(4.4)}$ | 60.7 $_{(0.1)}$ | 55.2 $_{(0.2)}$ | 35.9 $_{(3.1)}$ | 33.9 $_{(0.1)}$ | 33.5 $_{(0.3)}$ | 45.0 $_{(4.0)}$ | 69.4 $_{(0.1)}$ | 63.6 $_{(0.1)}$ |
| | | 5 | 51.6 $_{(3.2)}$ | 76.8 $_{(0.2)}$ | 62.9 $_{(0.1)}$ | 37.9 $_{(2.3)}$ | 38.3 $_{(0.2)}$ | 37.0 $_{(0.1)}$ | 53.0 $_{(3.6)}$ | 79.0 $_{(0.2)}$ | 70.4 $_{(0.3)}$ |
| | | 10 | 56.1 $_{(4.6)}$ | 81.3 $_{(0.1)}$ | 65.1 $_{(0.1)}$ | 38.4 $_{(3.8)}$ | 37.5 $_{(0.1)}$ | 40.0 $_{(0.2)}$ | 58.0 $_{(2.3)}$ | 83.4 $_{(0.1)}$ | 79.2 $_{(0.1)}$ |
| | $\mathcal{D}_{\text{FEED}}$ | 1 | 42.8 $_{(4.3)}$ | **49.7** $_{(1.0)}$ | **49.7** $_{(1.0)}$ | 27.6 $_{(2.4)}$ | **32.3** $_{(1.5)}$ | **32.3** $_{(1.3)}$ | **41.2** $_{(2.1)}$ | **56.8** $_{(1.8)}$ | **56.8** $_{(1.8)}$ |
| | | 2 | **54.8** $_{(3.0)}$ | 60.5 $_{(1.1)}$ | 54.8 $_{(0.7)}$ | **39.5** $_{(2.5)}$ | 32.6 $_{(1.2)}$ | 32.7 $_{(1.1)}$ | **53.8** $_{(2.3)}$ | 68.6 $_{(1.7)}$ | 63.5 $_{(1.3)}$ |
| | | 5 | **53.7** $_{(3.8)}$ | **77.9** $_{(0.8)}$ | 61.5 $_{(1.4)}$ | **39.2** $_{(2.0)}$ | 38.3 $_{(1.3)}$ | **39.4** $_{(1.0)}$ | **58.2** $_{(2.8)}$ | 78.8 $_{(1.6)}$ | **71.8** $_{(1.4)}$ |
| | | 10 | **58.0** $_{(3.4)}$ | **85.8** $_{(0.9)}$ | **67.8** $_{(1.2)}$ | **39.7** $_{(2.8)}$ | **39.0** $_{(1.0)}$ | **41.6** $_{(1.3)}$ | **59.8** $_{(3.1)}$ | **86.0** $_{(1.9)}$ | **83.4** $_{(2.0)}$ |

solely unplugging $\mathcal{H}_i \cup \mathcal{H}_j$ does not impact the LLM's performance. Formally, we check whether the following equation holds:

$$Y_{(\{\boldsymbol{x}_{n'}|\boldsymbol{x}_{n'} \in \mathcal{D}_{\text{IN}}\})} = \mathbf{1}_{|\mathcal{D}_{\text{IN}}|}|\text{unplug}(\mathcal{H}_i \cup \mathcal{H}_j); C = \mathcal{D}_{\text{IN}}, S = (Y_{(\{\boldsymbol{x}_{n'}|\boldsymbol{x}_{n'} \in \mathcal{D}_{\text{IN}}\})} = \mathbf{1}_{|\mathcal{D}_{\text{IN}}|}), \quad (9)$$

which determines whether plugging $\mathcal{H}_i \cup \mathcal{H}_j$ is unnecessary for maintaining the correct outputs to all inputs in $\mathcal{D}_{\text{IN}}$. If Eq. (9) holds, we create a new node $\mathcal{H}_i \cup \mathcal{H}_j$ and add it to $\mathscr{H}_k$, labeling it with a MAINTAIN signal. Otherwise, we add both $\mathcal{H}_i$ and $\mathcal{H}_j$ to $\mathscr{H}_k$. After this computation, we identify $\mathcal{H}_{\text{MAX}} = \arg\max_{\mathcal{H}. \in \mathscr{H}_k} |\mathcal{H}.|$ and label it with a MAINTAIN signal. Subsequently, we remove the nodes in $\mathscr{H}_k$ that lack MAINTAIN signals. This process continues until $\mathscr{H}.$ contains only one element, denoted as $\mathcal{H}_{\text{UNNECESSARY}} \in \mathscr{H}_K$. Finally, we calculate $\mathcal{D}_{\text{OUT}}$ as $\mathcal{D}_{\text{OUT}} = \mathcal{D}_{\text{IN}} - \mathcal{H}_{\text{UNNECESSARY}}$.

We provide an illustrated example of this process in Figure 2(b). However, since at each iteration, we need to run the checker for $O(\text{C}_{N_{\text{IN}}}^2)$ times (where C. denotes a combination operator), this becomes impractical. To this end, we develop an alternative algorithm. Specifically, at each $k$-th iteration, we remove the checked nodes (i.e., $\mathcal{H}_i$ and $\mathcal{H}_j$ from $\mathscr{H}_k$ (similar to the sufficiency tree search). Then, it requires $O(\log_2^{|\mathcal{D}_{\text{IN}}|})$ iterations to finish one round. Then, similarly, it requires $O(\log_2^{|\mathcal{D}_{\text{IN}}|})$ iterations to finish one round. To obtain a necessary set, we need to repeat the above process for multiple rounds. This process continues until there is no or only one element left in $\mathscr{H}_0$. When practical, we can set $R_{\text{MAX}}$ as the maximum number of rounds to approximate. Overall, this approximate algorithm requires $O(R_{\text{MAX}} \log_2^{|\mathcal{D}_{\text{IN}}|})$ iterations. We illustrate the above process in Figure 2(c).

**Theorem 1.** *If we successively apply the sufficiency tree search algorithm and the necessity tree search algorithm (either the original one or the alternative one) on $\mathcal{D}_{\text{TRAIN}}$ to obtain $\mathcal{D}_{\text{FEED}}$, then $\mathcal{D}_{\text{FEED}}$ is a parsimonious set of $\mathcal{D}_{\text{TRAIN}}$, namely $\mathcal{D}_{\text{FEED}}$ satisfying Definition 3.*

Theorem 1 tells that both our sufficiency tree search algorithm and necessity tree search algorithm can serve as an efficient filter to eliminate the insufficient or unnecessary portion from $\mathcal{D}_{\text{TRAIN}}$. We provide comprehensive descriptions of the tree search algorithms, along with the analysis and the proof of the above theorem in Appendix C.

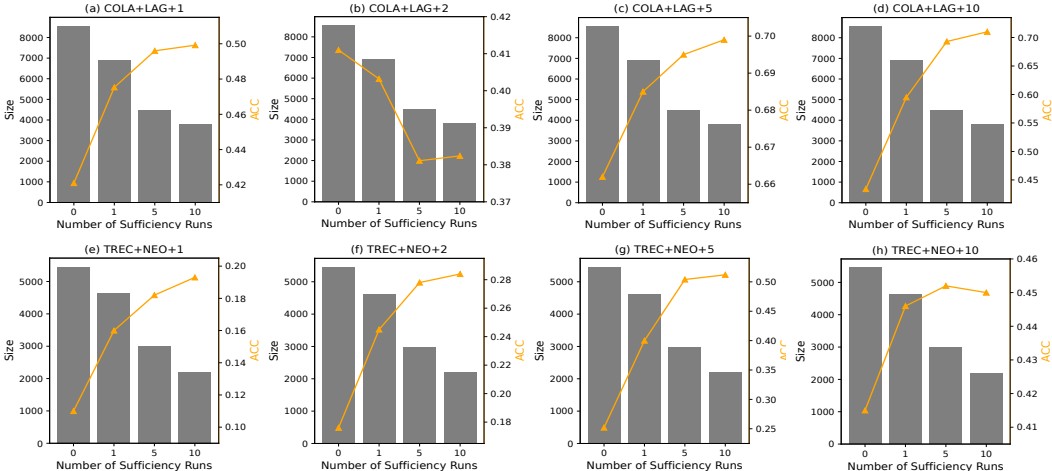

Figure 3: Performance comparisons of using different numbers of rounds of sufficient tree-search algorithm in terms of accuracy (denoted as ACC) and the size of the resulting sufficient set (denoted as Size). Each sub-figure is entitled with Dataset+LLM bases+n shots.

Due to the inherent limitation of context window size (up to 1024 word pieces), while practical, it is infeasible to directly finish the necessary check over all the instances in $\mathcal{D}_{\text{TRAIN}}$ when $|\mathcal{D}_{\text{TRAIN}}|$ is large. To overcome this challenge, we propose to incorporate a demonstration retriever between the sufficient set miner and the necessary set miner. In other words, we can first iteratively employ the sufficiency tree search for multiple rounds to obtain the sufficient set. Then, with each test input $x \in \mathcal{D}_{\text{TEST}}$, we run the retriever to get several demonstrations, and thus the necessary set miner only needs to filter out the unnecessary portions from the retrieved demonstrations. We illustrate the above justification in Appendix D.1.

# 5 DEPLOYING FEEDER INTO REAL-WORLD USE CASES

As previously discussed in Section 2, our primary focus lies in the in-context learning and fine-tuning settings, where our parsimonious set can represent and replace the whole training dataset. We mainly conducted our experiments using six text classification datasets: SST-2 (Socher et al., 2013), SST-5 (Socher et al., 2013), COLA (Warstadt et al., 2018), TREC (Voorhees & Tice, 2000), SUBJ (Pang & Lee, 2004), and FPB (Malo et al., 2014). These datasets cover a range of tasks from sentiment classification and linguistic analysis to textual entailment.

To evaluate the performance of our approach, we employed two GPT-2 variants (Radford et al., 2019): one with 335M parameters denoted as MED, and the other with 774M parameters denoted as LAR; one GPT-3 variant with 1.3B parameters denoted as NEO; and one Llama 2 variant (Touvron et al., 2023) with 7B parameters denoted as LLA as the LLM base (i.e., $\Psi_{\text{LLM}}(\cdot)$).

We also evaluate FEEDER on reasoning dataset GSM8K (Cobbe et al., 2021) and semantic-parsing dataset SMCALFlow (Andreas et al., 2020) with one GPT-3 variant with 6B parameters as the LLM, and report the results in Appendix D.3. All the code will be made available upon publication.

## 5.1 PERFORMANCE COMPARISONS FOR IN-CONTEXT LEARNING

We first evaluate the performance of FEEDER under the in-context learning setting. Due to the inherent limitation of context window size (up to 1024 word pieces), we assessed the LLM's accuracy in the n-shot setting across all datasets, with n values of one, two, five, and ten. We integrate FEEDER with existing demonstration retrievers, which involves first mining the parsimonious set using FEEDER and then retrieving relevant demonstrations from this set. We conducted an evaluation of FEEDER in conjunction with three commonly used retrievers: (i) RAN is the random retriever, which selects input demonstration randomly from the retrieval pool. (ii) SIM is the similarity-based retriever (Sorensen et al., 2022; Gonen et al., 2022), which selects relevant demonstrations in terms of the cosine similarity metric. (iii) DIV is the diversity-based retriever (Ye et al., 2022), which selects similar and diverse demonstrations in terms of maximal marginal relevance (Carbonell & Goldstein, 1998). When applying FEEDER, the retrieval pool is $\mathcal{D}_{\text{FEED}}$; otherwise, it is $\mathcal{D}_{\text{TRAIN}}$. Considering that the necessity tree search algorithm is time-consuming and either the sufficiency tree search algorithm

Table 2: Performance comparisons on text classification datasets for the fine-tuning setting. We report both the mean and variance of accuracy using four different seeds and four different permutations of n-shots. See Table 7 for more extended results on datasets COLA, SUBJ, and FPB.

| $\Psi_{\text{LLM}}(\cdot)$ | $\widetilde{\mathcal{D}}_{\text{TRAIN}}$ | $n$ | SUBJ | | | SST-5 | | | TREC | | |
|---|---|---|---|---|---|---|---|---|---|---|---|
| | | | RAN | SIM | DIV | RAN | SIM | DIV | RAN | SIM | DIV |
| NEO | $\mathcal{D}_{\text{TRAIN}}$ | 1 | 72.7 (5.7) | 91.0 (0.0) | 91.0 (0.0) | 17.3 (4.1) | 24.6 (0.0) | 24.6 (0.0) | 63.3 (5.2) | 79.5 (0.0) | 79.5 (0.0) |
| | | 2 | 74.1 (4.6) | 93.7 (0.0) | 92.1 (0.0) | 24.5 (3.2) | 25.8 (0.0) | 26.4 (0.0) | 63.5 (5.7) | 57.2 (0.0) | 51.4 (0.0) |
| | | 5 | 70.8 (5.1) | 93.3 (0.0) | 92.7 (0.0) | 23.6 (4.1) | 27.8 (0.0) | 27.3 (0.0) | 67.8 (4.7) | 66.6 (0.0) | 73.0 (0.0) |
| | | 10 | 89.2 (4.4) | 94.0 (0.0) | 91.6 (0.0) | 26.8 (2.9) | 26.5 (0.0) | 27.8 (0.0) | 66.9 (3.8) | 68.8 (0.0) | 72.4 (0.0) |
| | $\mathcal{D}_{\text{GOLD}}$ | 1 | **93.0** (4.3) | **93.5** (1.8) | **93.5** (1.8) | **19.5** (3.1) | **26.7** (2.0) | **26.7** (2.0) | **64.6** (3.2) | **80.6** (0.8) | **80.6** (0.8) |
| | | 2 | **96.1** (3.8) | **94.1** (1.3) | **92.6** (1.2) | **25.6** (2.7) | **26.4** (0.7) | **28.6** (0.8) | **64.2** (3.7) | **79.6** (0.7) | **80.3** (0.9) |
| | | 5 | **85.7** (3.5) | **94.7** (1.5) | **94.1** (1.1) | **27.4** (2.9) | **29.5** (1.8) | **29.7** (1.5) | **70.8** (3.2) | **78.2** (2.3) | **79.6** (1.9) |
| | | 10 | **90.5** (3.3) | **95.5** (1.3) | **95.6** (1.4) | **28.9** (2.0) | **28.6** (1.7) | **29.0** (1.6) | **69.7** (2.7) | **71.2** (1.7) | **76.7** (1.9) |

or the necessity tree search algorithm can serve as a data miner, we only operated the alternative necessity tree search algorithm for one round and ran the sufficiency tree search algorithm for five rounds. We conducted experiments with four different permutations for various n-shot scenarios and employed four different seeds for each experiment. In Appendix D.3, we also present additional results obtained using active learning techniques as retrievers. These retrievers encompass both the entropy-based retriever (Roy & McCallum, 2001) and the uncertainty-based retriever (Köksal et al., 2022). We offer comprehensive retriever descriptions in Appendix D.2.

Table 1 reports our main results, including both the mean and variance of accuracy. We evaluate the impact of the number of rounds running the sufficiency tree search algorithm and report the corresponding FEEDER's performance and the size of the sufficient set in Figure 3.

Our findings are summarized as follows:

FEEDER **is an effective data miner.** By combining Table 1 and Figure 3, one can observe that FEEDER allows us to retain nearly half of the training samples while still achieving superior or comparable performance in most scenarios. These results verify the effectiveness of FEEDER as an effective data filtering method.

FEEDER **is an effective input demonstration pre-filter, especially for large LLM bases and the random retriever.** Table 1 tells us that the performance of FEEDER is indeed influenced by both the choice of LLM and the retriever used in conjunction with it. Concretely, FEEDER tends to benefit from larger LLMs like LAR, NEO and LLA, with NEO and LLA showing especially significant improvements. This suggests that FEEDER leverages the capabilities of LLMs, including their reasoning skills and stored knowledge. Larger LLMs often possess more powerful capabilities, making them better suited for FEEDER. Additionally, FEEDER is also influenced by the choice of retrievers, with retrievers like RAN, FEEDER can provide significant improvements. This implies that FEEDER could effectively filter high-quality data from the training dataset.

FEEDER **operating on the set level, works well when the number of shots is large.** Table 1 shows that good performance of FEEDER often occurs when the number of shots is 5 or 10. This suggests that FEEDER measuring input demonstrations on the set level, can effectively address reciprocal inhibitions among data.

## 5.2 PERFORMANCE COMPARISONS FOR FINE-TUNING

We evaluate the performance of FEEDER under the fine-tuning setting. As introduced in Section 4 (and Appendix D.1), our approach can re-compute and store the sufficient set. Here, different from the in-context learning setting, we run our sufficiency tree search algorithm only for one round, and use the resulting sufficient set to fine-tune the LLM.

Table 2 reports our main results, including both the mean and variance of accuracy. We evaluate the performance changes of FEEDER in the context of using the different number of rounds of the sufficiency tree search algorithm and report the corresponding zero-shot results in Figure 4.

Our findings are summarized as follows:

FEEDER **can yield substantial enhancements when compared to fine-tuning with $\mathcal{D}_{\text{TRAIN}}$.** As shown in Table 2, FEEDER consistently delivers significant improvements over using $\mathcal{D}_{\text{TRAIN}}$. This demonstrates that achieving better performance is possible by using a small but high-quality dataset for fine-tuning, while also reducing computational expenses.

Table 3: Performance comparisons between using different LLM bases as the miner for acquiring a sufficient set on COLA dataset with different retrievers under a 1, 2, 5-shot in-context learning setting.

| $\mathcal{D}_{\text{FEED}}$ | $n$ | MED | | | LAR | | | NEO | | |
|---|---|---|---|---|---|---|---|---|---|---|
| | | RAN | SIM | DIV | RAN | SIM | DIV | RAN | SIM | DIV |
| MED | | 22.7 (5.7) | 31.0 (1.3) | 34.0 (1.4) | 28.3 (4.1) | 34.6 (1.8) | 34.6 (1.2) | 29.4 (5.2) | 29.5 (1.9) | 30.5 (1.8) |
| LAR | 1 | 24.1 (5.6) | 33.7 (1.4) | 36.1 (0.8) | 24.5 (3.2) | 35.8 (1.2) | 36.4 (1.0) | 23.5 (5.7) | 27.2 (2.0) | 31.4 (1.8) |
| NEO | | 29.4 (4.6) | 35.1 (1.5) | 35.1 (1.5) | 29.6 (3.8) | 35.1 (1.1) | 35.1 (1.1) | 28.3 (5.4) | 34.8 (1.3) | 34.8 (1.3) |
| MED | | 29.2 (4.4) | 38.9 (1.8) | 31.6 (2.0) | 34.8 (3.9) | 36.5 (2.3) | 27.8 (1.8) | 66.9 (3.8) | 63.8 (1.0) | 62.4 (1.3) |
| LAR | 2 | 30.2 (4.3) | 41.3 (1.8) | 32.2 (1.8) | 33.9 (3.1) | 37.4 (2.0) | 30.6 (2.0) | 65.6 (3.2) | 63.6 (0.8) | 63.5 (0.8) |
| NEO | | 31.1 (2.2) | 41.7 (1.2) | 34.9 (1.2) | 36.6 (3.5) | 37.0 (2.8) | 34.6 (2.0) | 69.3 (3.7) | 64.7 (1.4) | 64.7 (1.6) |
| MED | | 64.2 (4.4) | 56.9 (0.6) | 51.6 (0.4) | 66.8 (2.9) | 67.4 (1.0) | 67.8 (2.0) | 66.9 (3.8) | 65.8 (2.3) | 62.6 (1.6) |
| LAR | 5 | 65.3 (4.3) | 56.3 (1.8) | 52.4 (1.8) | 68.8 (3.1) | 67.7 (2.0) | 67.7 (2.0) | 67.7 (3.2) | 66.0 (0.8) | 64.3 (0.8) |
| NEO | | 65.2 (2.0) | 57.3 (1.2) | 54.6 (1.7) | 69.2 (3.3) | 68.6 (1.6) | 66.6 (1.7) | 68.7 (3.2) | 67.2 (2.4) | 65.8 (1.8) |

FEEDER **performs optimally when the sufficiency tree search algorithm runs for just one round.** Figure 4 illustrates the influence of employing varying numbers of rounds for constructing the sufficient set for NEO's fine-tuning. For ease of comparison, we also include the results of fine-tuning NEO on $\mathcal{D}_{\text{TRAIN}}$ with the blue lines. These findings suggest that fine-tuning with a smaller dataset can improve final performance, but an excessively small dataset may not yield the desired results. Similar results would be found in (Chen et al., 2023).

## 5.3 SCALING UP FEEDER TO LLMS AND DYNAMIC DATA

**Scaling Up** FEEDER **to Larger LLMs.** As the LLM scales up in size, the execution of our sufficiency and necessity tree search algorithms can become exceedingly time-consuming. In response, we suggest employing a smaller LLM to generate a sufficient set, which can then be stored and utilized by the larger LLM. To assess the viability of this approach, we conducted an experiment to compare the performance of using MED, LAR and NEO as the LLM to get the sufficient set and use it as the retrieval pool to get demonstrations for the LLM MED, LAR and NEO, and we summarize the results in Table 3. From the table, we can observe that there are slight differences in using different LLMs as the miner to compute $\mathcal{D}_{\text{FEED}}$. This observation suggests the potential feasibility of employing a more compact LLM for mining high-quality demonstrations to benefit a larger LLM. However, this inference may not hold true when comparing a smaller LLM fine-tuned on domain-specific data to a larger LLM fine-tuned on general knowledge, due to their dissimilar knowledge bases and inferential capabilities.

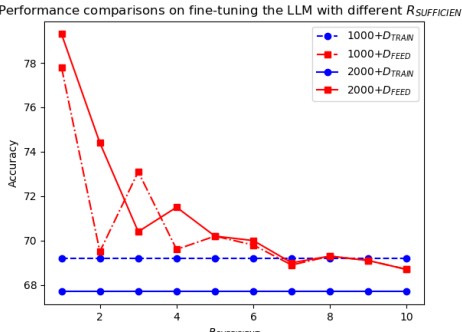

Figure 4: Performance comparisons on fine-tuning NEO on the sufficient set with different rounds of running our sufficiency tree search algorithm (i.e., $R_{\text{SUFFICIENT}}$). Our evaluation operates on COLA dataset in the zero-shot setting after fine-tuning on 1000 and 2000 batches.

**Scaling Up** FEEDER **to Dynamic Data.** Notice that many real-world datasets for training are temporal and require frequent updates. Re-running FEEDER for all the samples is super time-consuming. Instead, we treat the unchanged part of plug-and-play the parsimonious set and the LLM as the whole, as a new "LLM model", and therefore, we only apply FEEDER to compute incremental data for the changed part (including newly added and modified data points). We illustrate the above process in Appendix D.1.

In Appendix D.5, we present a time complexity evaluation for FEEDER. Additionally, in Appendix E, we offer a case study based on some artificial cases to provide a detailed illustration of FEEDER's functionality.

## 6 CONCLUSION AND FUTURE WORK

In this paper, we introduce a novel LLM framework FEEDER, addressing the identification of highly effective data and the efficient approach to discovering them. Our experimental findings demonstrate FEEDER could offer substantial benefits across various LLMs with the collaborations with different retrievers. In the future, it would be interesting to explore additional use cases for FEEDER in data safety and data management.

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

# A    CONNECTIONS TO EXISTING APPROACHES

## A.1    CONNECTIONS TO CAUSALITY

Concepts of sufficiency and necessity have a broad application scope, especially in causality (Pearl, 1980; 2009), where sufficiency and necessity are proposed to define the causal relationship between two binary variables. Let $X$ and $Y$ denote a pair of variables. Then, the probability of sufficiency measures the capacity of setting $X = \mathtt{true}$ to produce $Y = \mathtt{true}$, while the probability of necessity measures the changing the value of $X$ from $X = \mathtt{true}$ to $X = \mathtt{false}$ would cause the value of $Y$ changing from $Y = \mathtt{true}$ to $Y = \mathtt{false}$.

In this paper, we adopt the concepts of sufficiency and necessity in the context of demonstration mining, where we investigate whether prompting certain data points is sufficient or necessary for the given LLM to generate the correct answers for the input questions. For this purpose, we introduce the plugging-in operation, denoted as $\mathtt{plug}(\cdot)$, to examine the sufficiency and the unplugging operation, denoted as $\mathtt{unplug}(\cdot)$, to examine the necessity. Both of the above operations are similar to the do operation in the causality, denoted as $\mathtt{do}(\cdot)$, which indicates that the system operates under the condition that certain variables are controlled by external forces.

To be more specific, in our setting, the external force can be explained as: we can choose to either plug-in or unplug some data points to change what is already plugged for the LLM. Our intuition is similar to the counterfactual idea in the casualty, which investigates what if some variables are set with some different values. In our case, we study what if the plugged-in data includes some data points that are different from the historical (a.k.a., factual) setting. One significant difference between ours and the counterfactual setting in casualty is that we do not need to estimate the "counterfactual" situations (namely setting plugged-in data with different values), as we can directly do evaluations.

## A.2    CONNECTIONS TO DEMONSTRATION SELECTION

As LLMs have shown great potential in few-shot inference, a core challenge is how to select "shots". One principal solution is to select some training examples as the extra input, which are named as demonstrations or prompts (Levy et al., 2022; Liu et al., 2021; Dong et al., 2022). One implicit assumption here is that the training set is a support set (Yosida, 2012) of test samples. Previous investigations (Wang et al., 2022; Rubin et al., 2021) show that plugging in similar training examples can improve the performance of LLMs on test instances. However, as pointed out in (Levy et al., 2022), these methods only measure each data point in isolation, instead of considering the data points as a whole. In other words, a retriever based on the similarity metric, only operates on the instance level and could select two same data points together as the extra input, which are definitively redundant. To address this issue, a recent paper (Levy et al., 2022) proposes to consider the diversity among the data points, to avoid the case where too "similar" data points are selected together.

In this paper, our main insight is that the quality of demonstrations would be aware of the LLM in use. In other words, a high-quality demonstration for one LLM could be a low-quality demonstration for another LLM. In light of this, we propose sufficiency and necessity as the new set-level metrics. Our method conveys the following advantages. Firstly, compared to the similarity and diversity metrics, sufficiency and necessity measure the quality of data points depending on the LLM in use. Secondly, compared to instance-level metrics, our proposed sufficiency and necessity can be extended to the set level, and thus can consider the data points as a whole. In our framework, the "similarity" can be translated as "sufficiency", meaning that plugging in the data points can improve the performance of the LLM, while the "diversity" can be translated as "necessity", indicating that each data point should play an irreplaceable role.

## A.3    CONNECTIONS TO ACTIVE LEARNING AND CORE-SET SELECTION

Core-set selection (Feldman, 2020; Guo et al., 2022), aiming to select a subset of the most informative training samples, is a long-standing learning problem that can benefit many downstream tasks such as active learning. One prior work (Dor et al., 2020) has summarized and evaluated existing state-of-the-art active learning approaches for BERT (Devlin et al., 2018), including random sampling, uncertainty-sampling (using entropy metric) (Lewis, 1995; Gal & Ghahramani, 2016) and diversity sampling (using diversity metric) (Gissin & Shalev-Shwartz, 2019). Different from active learning,

FEEDER selects core sets (called parsimonious sets) that can be either used as the extra input contexts or used to train the LLM. And, the "uncertainty" could select the data points with max entropy, independent of the LLM. In contrast, "informative training samples" in our framework are defined as those training samples that can benefit the LLM's performance on a specific task.

# B A FAMILY OF ANALYSIS ON DATA RELATIONSHIPS

We begin by describing some key notations used in the paper.

**Notation.** Let $X, C$ denote variables for the input and the context (i.e., previously plugged-in demon stations). We use $Y$, a boolean variable, to denote whether the output to the input is correct. Concretely, we use $Y_{\boldsymbol{x}} = 1$ to denote $Y = 1|X = \boldsymbol{x}$, meaning that the LLM generates the correct output to the input $\boldsymbol{x}$. Similarly, $Y_{\boldsymbol{x}} = 0$, equivalent to $Y = 0|X = \boldsymbol{x}$, indicates that the LLM produces the incorrect output to $\boldsymbol{x}$. For clarity, we introduce $S$, a variable to record the original status of the LLM before *new* plug-in and unplug operations (denoted as $\texttt{plug}(\cdot)$ and $\texttt{unplug}(\cdot)$ respectively), e.g., $C = ((\boldsymbol{x}, \boldsymbol{y})), S = (Y_{\boldsymbol{x}} = 1)$ means that without plugging-in any new data or unplugging any plugged-in data, the plugged-in data is $(\boldsymbol{x}, \boldsymbol{y})$ and the LLM's performance is $Y_{\boldsymbol{x}} = 1$.

## B.1 DATA RELATIONSHIPS ON INSTANCE LEVEL

In this subsection, we consider two instances, denoted as $(\boldsymbol{x}_n, \boldsymbol{y}_n)$ and $(\boldsymbol{x}_m, \boldsymbol{y}_m)$.

*Sufficiency* relationship is proposed to evaluate whether plugging-in one data point is *sufficient* to enable the LLM to generate the correct output to the other one. Formally, we have:

**Definition 4 (Sufficient Instance).** *Given tuple* $(X, Y, C, S)$*, we say that the data point* $(\boldsymbol{x}_n, \boldsymbol{y}_n)$ *is a sufficient instance for* $(\boldsymbol{x}_m, \boldsymbol{y}_m)$ *(namely plugging-in* $(\boldsymbol{x}_n, \boldsymbol{y}_n)$ *is sufficient to get the correct output to* $\boldsymbol{x}_m$*), if the following equation holds:*

$$Y_{\boldsymbol{x}_m} = 1|\texttt{plug}((\boldsymbol{x}_n, \boldsymbol{y}_n)); C = \emptyset, S, \tag{10}$$

*where $S$ can be any value. It indicates that before plugging-in* $(\boldsymbol{x}_n, \boldsymbol{y}_n)$*, there is no plugged-in data as contexts (i.e., $C = \emptyset$), and when we plug-in* $(\boldsymbol{x}_n, \boldsymbol{y}_n)$ *(i.e., $\texttt{plug}((\boldsymbol{x}_n, \boldsymbol{y}_n))$), then it results in the LLM's output to $\boldsymbol{x}_m$ is correct (i.e., $Y_{\boldsymbol{x}_m} = 1$).*

**Example A1.** Let $\boldsymbol{x}_m, \boldsymbol{x}_n$ be *Which country does Sherlock Holmes live?* and *Which city does Sherlock Holmes live?* Suppose that without any plugged-in data, the LLM generates wrong answers for the above two questions, e.g., *Sherlock Holmes lives in the United States* for $\boldsymbol{x}_m$ and *Sherlock Holmes lives in New York* for $\boldsymbol{x}_n$. Then, after telling the LLM the correct answer of $\boldsymbol{x}_n$ (e.g., $\boldsymbol{y}_n$ is *Sherlock Holmes lives in London*), then the LLM can infer the correct answer of $\boldsymbol{x}_m$ (e.g., $\boldsymbol{y}_m$ is *Sherlock Holmes lives in the United Kingdom*). In this case, the LLM is using the city where Sherlock Holmes lives to infer the country in which he lives.

*Necessity* relationship is proposed to evaluate whether it is *necessary* to keep one plugged-in data point in terms of maintaining the correct output to the other one. Formally, we have:

**Definition 5 (Necessary Instance).** *Given tuple* $(X, Y, C, S)$*, we say that the data point* $(\boldsymbol{x}_n, \boldsymbol{y}_n)$ *is a necessary instance for* $(\boldsymbol{x}_m, \boldsymbol{y}_m)$ *(namely plugging-in* $(\boldsymbol{x}_n, \boldsymbol{y}_n)$ *is necessary to get the correct output to $\boldsymbol{x}_m$), if the following equation holds:*

$$Y_{\boldsymbol{x}_m} = 0|\texttt{unplug}((\boldsymbol{x}_n, \boldsymbol{y}_n)); C = ((\boldsymbol{x}_n, \boldsymbol{y}_n)), S = (Y_{\boldsymbol{x}_m} = 1). \tag{11}$$

*It means that before unplugging* $(\boldsymbol{x}_n, \boldsymbol{y}_n)$*, there is plugged-in data as the context* $(\boldsymbol{x}_n, \boldsymbol{y}_n)$ *(i.e., $C = ((\boldsymbol{x}_n, \boldsymbol{y}_n))$) and the LLM's answer to $\boldsymbol{x}_m$ is correct (i.e., $S = (Y_{\boldsymbol{x}_m} = 1)$), and when we unplug* $(\boldsymbol{x}_n, \boldsymbol{y}_n)$ *(i.e., $\texttt{unplug}((\boldsymbol{x}_n, \boldsymbol{y}_n))$), then it would cause the LLM offering an incorrect output (i.e., $Y_{\boldsymbol{x}_m} = 0$).*

**Example A2.** Let $\boldsymbol{x}_m, \boldsymbol{x}_n$ be *Which city does Sherlock Holmes live?* and *What is the detailed address of Sherlock Holmes lives?* Suppose that the LLM knows nothing about Sherlock Holmes without previous plugged-in $(\boldsymbol{x}_n, \boldsymbol{y}_n)$ (where $\boldsymbol{y}_n$ is *221B Baker Street, London*). After plugging-in $(\boldsymbol{x}_n, \boldsymbol{y}_n)$, the LLM can produce a correct output $\boldsymbol{y}_m$ (where $\boldsymbol{y}_m$ is *Sherlock Holmes live in London*) to $\boldsymbol{x}_m$. If unplugging $(\boldsymbol{x}_n, \boldsymbol{y}_n)$, the LLM would produce an incorrect output to $\boldsymbol{x}_m$, e.g., *Sherlock Holmes live in New York*.

Ideally, when we aim to ensure the LLM's performance, a principle solution is to extract data points that are both sufficient and necessary, to guarantee accuracy meanwhile avoiding data redundancy.

**Definition 6 (Sufficient and Necessary Instance).** *Given tuple $(X, Y, C)$, we say that data point $(\boldsymbol{x}_n, \boldsymbol{y}_n)$ is a sufficient and necessary instance for $(\boldsymbol{x}_m, \boldsymbol{y}_m)$ (namely plugging-in $(\boldsymbol{x}_n, \boldsymbol{y}_n)$ is both sufficient and necessary to get the correct output to $\boldsymbol{x}_m$), if the following equation holds:*

$$\left(Y_{\boldsymbol{x}_m} = 1 | \mathtt{plug}((\boldsymbol{x}_n, \boldsymbol{y}_n)); C = \emptyset\right) \wedge \left(Y_{\boldsymbol{x}_m} = 0 | \mathtt{unplug}((\boldsymbol{x}_n, \boldsymbol{y}_n)); C = ((\boldsymbol{x}_n, \boldsymbol{y}_n))\right), \quad (12)$$

*which indicates that plugging-in data point $(\boldsymbol{x}_n, \boldsymbol{y}_n)$ can respond to the LLM's answering $\boldsymbol{x}_m$ in both ways. We omit $S$ here, because we can derive the original status of the necessary instance based on the condition of the sufficiency instance.*

Although neither of the above two quantities (i.e., sufficiency and necessity) is sufficient for determining the other one, they are not entirely independent, as shown in the following lemma.

**Lemma 1.** *Supposing that we only consider using the data point $(\boldsymbol{x}_n, \boldsymbol{y}_n)$ as the plug-in data, and only care about the LLM's performance regarding the input question $\boldsymbol{x}_m$, then overall there are only two situations here: (i) $(\boldsymbol{x}_n, \boldsymbol{y}_n)$ is plugged-in, and (ii) $(\boldsymbol{x}_n, \boldsymbol{y}_n)$ is not plugged-in. Based on the above assumption, we re-write (i) as plugging-in $(\boldsymbol{x}_n, \boldsymbol{y}_n)$ when there is no plugged-in data (i.e., $\mathtt{plug}((\boldsymbol{x}_n, \boldsymbol{y}_n)); C = \emptyset$, and re-write (ii) as unplugging $(\boldsymbol{x}_n, \boldsymbol{y}_n)$ when there is plugged-in data $(\boldsymbol{x}_n, \boldsymbol{y}_n)$ (i.e., $\mathtt{unplug}((\boldsymbol{x}_n, \boldsymbol{y}_n)); C = ((\boldsymbol{x}_n, \boldsymbol{y}_n))$). For simplicity, we use $C^*$ and $C$ to denote (i) and (ii) respectively; and we use $Y^*$ and $Y$ to denote $Y_{\boldsymbol{x}_1} = 1$ and $Y_{\boldsymbol{x}_1} = 0$. Then, we have: $C^* \vee C = \mathtt{true}, C^* \wedge C = \mathtt{false}, Y^* \vee Y = \mathtt{true}, Y^* \wedge Y = \mathtt{false}.$*

*We define* PS *as the probability of being sufficient instance as:*

$$\mathtt{PS} := \mathtt{Pr}\left(Y_{\boldsymbol{x}_m} = 1 | \mathtt{plug}((\boldsymbol{x}_n, \boldsymbol{y}_n)); C = \emptyset\right) = \mathtt{Pr}(Y^* | C^*). \quad (13)$$

*We define* PN *as the probability of being necessary instance as:*

$$\mathtt{PN} := \mathtt{Pr}\left(Y_{\boldsymbol{x}_m} = 0 | \mathtt{unplug}((\boldsymbol{x}_n, \boldsymbol{y}_n)); C = ((\boldsymbol{x}_n, \boldsymbol{y}_n))\right) = \mathtt{Pr}(Y | C). \quad (14)$$

*We further define* PNS *as the probability of being sufficient and necessary instance as:*

$$\mathtt{PNS} := \mathtt{Pr}(Y^* | C^*, Y | C). \quad (15)$$

*Then,* PS*,* PN*,* PSN *satisfy the following relationship:*

$$\mathtt{PSN} = \mathtt{Pr}(Y, C) \cdot \mathtt{PS} + \mathtt{Pr}(Y^*, C^*) \cdot \mathtt{PN}. \quad (16)$$

*Proof.* From the above description of $Y^*, Y, C^*, C$, we can write:

$$\begin{aligned}
Y^* | C^* \wedge Y | C &= (Y^* | C^* \wedge Y | C) \wedge (C \vee C^*) \\
&= (Y^* | C^* \wedge Y \wedge C) \vee (Y | C \wedge Y^* \wedge C^*).
\end{aligned} \quad (17)$$

Taking probabilities on both sides and using the disjointedness of $C^*$ and $C$, we have:

$$\begin{aligned}
\mathtt{PSN} &= \mathtt{Pr}(Y^* | C^*, Y | C) = \mathtt{Pr}(Y | C, Y^*, C^*) + \mathtt{Pr}(Y^* | C^*, Y, C) \\
&= \mathtt{Pr}(Y, C) \cdot \mathtt{PS} + \mathtt{Pr}(Y^*, C^*) \cdot \mathtt{PN}.
\end{aligned} \quad (18)$$

$\square$

## B.2 DATA RELATIONSHIPS ON SET LEVEL

In this subsection, we first extend Definitions 4 and 5 to the set level as:

**Definition 7 (Sufficient Set).** *Given tuple $(X, Y, C, S)$, we say that the input set $\mathcal{D}_{\mathtt{IN}}$ is a sufficient set for output set $\mathcal{D}_{\mathtt{OUT}}$ (namely plugging-in $\mathcal{D}_{\mathtt{IN}}$ is sufficient to get the correct output to **any** data point in $\mathcal{D}_{\mathtt{OUT}}$), if the following equation holds:*

$$Y_{(\{\boldsymbol{x}_n | \boldsymbol{x}_n \in \mathcal{D}_{\mathtt{OUT}}\})} = \mathbf{1}_{N_{\mathtt{OUT}}} | \mathtt{plug}(\mathcal{D}_{\mathtt{IN}}); C = \emptyset, S, \quad (19)$$

*where $S$ can be any value and $N_{\mathtt{OUT}} = |\mathcal{D}_{\mathtt{OUT}}|$. It indicates that when plugging-in $\mathcal{D}_{\mathtt{IN}}$ (i.e., $\mathtt{plug}(\mathcal{D}_{\mathtt{IN}})$), then it would guarantee that the LLM's output to any input question in $\mathcal{D}_{\mathtt{OUT}}$ is correct (i.e., $Y_{(\{\boldsymbol{x}_n | \boldsymbol{x}_n \in \mathcal{D}_{\mathtt{OUT}}\})} = \mathbf{1}_{N_{\mathtt{OUT}}}$).*

**Definition 8 (Necessary Set).** *Given tuple $(X, Y, C, S)$, we say that the input set $\mathcal{D}_{\text{IN}}$ is a necessary set for output set $\mathcal{D}_{\text{OUT}}$ (namely plugging-in **any** subset of $\mathcal{D}_{\text{IN}}$, denoted as $\mathcal{D}'_{\text{IN}}$, is necessary to get the correct output to **at least one** data point in $\mathcal{D}_{\text{OUT}}$), if the following equation holds:*

$$Y_{(\{\boldsymbol{x}_n | \boldsymbol{x}_n \in \mathcal{D}_{\text{OUT}}\})} \neq \mathbf{1}_{N_{\text{OUT}}} | \texttt{unplug}(\mathcal{D}'_{\text{IN}}); C = \mathcal{D}_{\text{IN}}, S = (Y_{(\{\boldsymbol{x}_n | \boldsymbol{x}_n \in \mathcal{D}_{\text{OUT}}\})} = \mathbf{1}_{N_{\text{OUT}}}), \quad (20)$$

*where $N_{\text{OUT}} = |\mathcal{D}_{\text{OUT}}|$. It means that before unplugging any subset of $\mathcal{D}_{\text{IN}}$, there is plugged-in data $\mathcal{D}_{\text{IN}}$ (i.e., $C = \mathcal{D}_{\text{IN}}$) and the LLM's output to any input in $\mathcal{D}_{\text{OUT}}$ is correct (i.e., $S = (Y_{(\{\boldsymbol{x}_n | \boldsymbol{x}_n \in \mathcal{D}_{\text{OUT}}\})} = \mathbf{1}_{N_{\text{OUT}}})$), and when we unplug any subset of $\mathcal{D}_{\text{IN}}$ (i.e., $\texttt{unplug}(\mathcal{D}'_{\text{IN}})$), then it would cause the LLM's output to at least one input in $\mathcal{D}_{\text{OUT}}$ being incorrect (i.e., $Y_{(\{\boldsymbol{x}_n | \boldsymbol{x}_n \in \mathcal{D}_{\text{OUT}}\})} \neq \mathbf{1}_{N_{\text{OUT}}}$).*

From the above descriptions, one can see that when we say one set is a sufficient set, we require that the overall set of data points is sufficient; when we say one set is a necessary set, we require that each data point in the set is necessary.

**Example A3.** *Let $\mathcal{D}_{\text{OUT}} = \{(\boldsymbol{x}_m, \boldsymbol{y}_m)\}$ and $\mathcal{D}_{\text{IN}} = \{(\boldsymbol{x}_i, \boldsymbol{y}_i), (\boldsymbol{x}_j, \boldsymbol{y}_j)\}$. We assign $\boldsymbol{x}_m$ and $\boldsymbol{y}_m$ as Which country does Sherlock Holmes live? and Sherlock Holmes lives in the United Kingdom. Let $\boldsymbol{x}_i$ and $\boldsymbol{y}_i$ denote Which street does Sherlock Holmes live? and Baker street. We assign $\boldsymbol{x}_j$ and $\boldsymbol{y}_j$ as Where is Baker street? and Bake street is located in London. Supposing that the LLM does not know that Bake street is located in the United Kingdom, then solely plugging-in either $(\boldsymbol{x}_i, \boldsymbol{y}_i)$ or $(\boldsymbol{x}_j, \boldsymbol{y}_j)$ is not sufficient for the LLM to get the right answer $\boldsymbol{y}_m$ to the input question $\boldsymbol{x}_m$. In this regard, it is easy to derive that $\mathcal{D}_{\text{IN}}$ is both sufficient and necessary plug-in set for $\mathcal{D}_{\text{OUT}}$: (i) plugging-in $\mathcal{D}_{\text{IN}}$ is sufficient to maintain the right answer for $\mathcal{D}_{\text{OUT}}$; and (ii) unplugging any subset of $\mathcal{D}_{\text{IN}}$ can not maintain the right answer for $\mathcal{D}_{\text{OUT}}$.*

Then, we investigate the following problem: how to define a subset in the given dataset $\mathcal{D}$ that is both sufficient and necessary to represent $\mathcal{D}$. Formally, we have:

**Definition 9 (Parsimonious Set).** *Given tuple $(X, Y, C, S)$ and dataset $\mathcal{D} = \{(\boldsymbol{x}_n, \boldsymbol{y}_n)\}_{n=1}^N$, we say that $\mathcal{D}$ can be partitioned into two parts: one is a parsimonious set, denoted as $\mathcal{D}_{\text{FEED}}$, and the other part is denoted as $\overline{\mathcal{D}}_{\text{FEED}}$, if the following conditions are satisfied:*

*(i) $Y_{(\boldsymbol{x}_1..., \boldsymbol{x}_N)} = \mathbf{1}_N | \texttt{plug}(\mathcal{D}_{\text{FEED}}); C = \emptyset, S$ holds, where $S$ can be any value, and $\mathbf{1}_N$ and $\mathbf{0}_N$ denotes $N$-dimensional vectors whose elements are all 1s and 0s. It indicates that plugging-in $\mathcal{D}_{\text{FEED}}$ alone is sufficient for maintaining $Y_{(\boldsymbol{x}_1..., \boldsymbol{x}_N)} = \mathbf{1}_N$.*

*(ii) $Y_{(\boldsymbol{x}_1..., \boldsymbol{x}_N)} \neq \mathbf{1}_N | \texttt{unplug}(\mathcal{D}'_{\text{FEED}} \cup \overline{\mathcal{D}}_{\text{FEED}}); C = \mathcal{D}, S = (Y_{(\boldsymbol{x}_1..., \boldsymbol{x}_N)} = \mathbf{1}_N))$ holds for any subset of $\mathcal{D}_{\text{FEED}}$ (denoted as $\mathcal{D}'_{\text{FEED}}$). It indicates that if $\overline{\mathcal{D}}_{\text{FEED}}$ would be unplugged, then unplugging any data point in $\mathcal{D}_{\text{FEED}}$ would make the plugged-in data not sufficient to keep the output accurate for all the inputs. Namely, plugging-in $\mathcal{D}_{\text{FEED}}$ is necessary for maintaining $Y_{(\boldsymbol{x}_1..., \boldsymbol{x}_N)} = \mathbf{1}_N$.*

**Example A4.** *If we merge $\mathcal{D}_{\text{IN}}$ and $\mathcal{D}_{\text{OUT}}$ exemplified in Example A3 into one set $\mathcal{D}$, namely let $\mathcal{D} = \mathcal{D}_{\text{IN}} \cup \mathcal{D}_{\text{OUT}}$, then in this case, it is easy to derive that $\mathcal{D}_{\text{IN}}$ is a parsimonious set for $\mathcal{D}$.*

**Theorem 2.** *For the given dataset $\mathcal{D}$, $\mathcal{D}_{\text{FEED}}$ is a parsimonious set of $\mathcal{D}$, iff $\mathcal{D}_{\text{FEED}}$ is a sufficient set of $\mathcal{D}$ and $\mathcal{D}_{\text{FEED}}$ is a necessary set of $\mathcal{D}$.*

*Proof.* Firstly, it is simple to show that condition (i) in Definition 9 is equivalent to the sufficiency in Definition 7. Both of them can reach that

$$Y_{(\{\boldsymbol{x}_n | \boldsymbol{x}_n \in \mathcal{D}\})} = \mathbf{1}_N | \texttt{plug}(\mathcal{D}_{\text{FEED}}); Z = \emptyset, \quad (21)$$

which indicates that plugging-in $\mathcal{D}_{\text{FEED}}$ is sufficient to guarantee the LLM's output to *any* input in $\mathcal{D}$.

Secondly, we re-write condition (ii) in Definition 9 as:

$$Y_{(\{\boldsymbol{x}_n | \boldsymbol{x}_n \in \mathcal{D}\})} \neq \mathbf{1}_N | \texttt{unplug}(\mathcal{D}'_{\text{FEED}}); Z = \mathcal{D}_{\text{FEED}}, S = (Y_{(\{\boldsymbol{x}_n | \boldsymbol{x}_n \in \mathcal{D}\})} = \mathbf{1}_N), \quad (22)$$

where $\mathcal{D}'_{\text{FEED}}$ can be any subset of $\mathcal{D}_{\text{FEED}}$. It means that any data point is necessary to maintain the LLM's correct outputs to the inputs in $\mathcal{D}$, namely $\mathcal{D}_{\text{FEED}}$ is a necessary set for $\mathcal{D}$.

Similarly, one can directly combine sufficiency and necessity to derive conditions (i) and (ii). $\square$

---

**Algorithm 1:** Sufficiency Tree Search Algorithm in FEEDER

---

**Input:** Input dataset $\mathcal{D}_{\text{IN}} = \{(\boldsymbol{x}_n, \boldsymbol{y}_n)\}_{n=1}^{N_{\text{IN}}}$.
**Output:** A sufficient set of $\mathcal{D}_{\text{IN}}$, denoted as $\mathcal{D}_{\text{OUT}}$.
Initialize $k = 1$ and $\mathscr{W}_0 = \{\mathcal{W}_n = \{(\boldsymbol{x}_n, \boldsymbol{y}_n)\}|(\boldsymbol{x}_n, \boldsymbol{y}_n) \in \mathcal{D}_{\text{IN}}\}$.
**repeat**
Initialize $k = 1$ and $\mathscr{W}_0 = \{\mathcal{W}_n = \{(\boldsymbol{x}_n, \boldsymbol{y}_n)\}|(\boldsymbol{x}_n, \boldsymbol{y}_n) \in \mathcal{D}_{\text{IN}}\}$.
**foreach** *pair* $(\mathcal{W}_i, \mathcal{W}_j)$ *where* $\mathcal{W}_i, \mathcal{W}_j \in \mathscr{W}_{k-1}$ **do**
    Check $Y_{(\{\boldsymbol{x}_n|\boldsymbol{x}_n \in \mathcal{W}_j\})} = \mathbf{1}_{|\mathcal{W}_j|}|\text{plug}(\mathcal{W}_i); C = \emptyset$ (a).
    Check $Y_{(\{\boldsymbol{x}_n|\boldsymbol{x}_n \in \mathcal{W}_i\})} = \mathbf{1}_{|\mathcal{W}_i|}|\text{plug}(\mathcal{W}_j); C = \emptyset$ (b).
    **Case I** (both (a) and (b) hold), if $|\mathcal{W}_i| \geq |\mathcal{W}_j|$, append $\mathcal{W}_j$ to $\mathscr{W}_k$; otherwise, append $\mathcal{W}_i$ to
    $\mathscr{W}_k$.
    **Case II** (either one of (a) and (b) holds), if (a) holds, append $\mathcal{W}_i$ to $\mathscr{W}_k$; otherwise, append
    $\mathcal{W}_j$ to $\mathscr{W}_k$.
    **Case III** (neither (a) nor (b) holds), append $\mathcal{W}_i \cup \mathcal{W}_j$ to $\mathscr{W}_k$.
    Remove $\mathcal{W}_i, \mathcal{W}_j$ from $\mathscr{W}_{k-1}$, i.e., $\mathscr{W}_{k-1} = \mathscr{W}_{k-1} - \{\mathcal{W}_i, \mathcal{W}_j\}$.
**end**
**if** $|\mathscr{W}_{k-1}| = 1$ **then**
    Append only element in $\mathscr{W}_{k-1}$ to $\mathscr{W}_k$.
**end**
Grow tree from bottom to top via $k = k + 1$.
**until** $|\mathscr{W}_k| = 1$, and we assume the current iteration is $K$.
Let $\mathcal{W}_{\text{SUFFICIENT}}$ denote only one element (i.e. the root node) in $\mathscr{W}_K$.
Assign $\mathcal{D}_{\text{OUT}}$ as $\mathcal{W}_{\text{SUFFICIENT}}$, i.e., $\mathcal{D}_{\text{OUT}} = \mathcal{W}_{\text{SUFFICIENT}}$.

---

## C  FROM THEORY TO PRACTICE

Aligning with notations introduced in Appendix B.2, we use $\mathcal{D}_{\text{IN}} = \{(\boldsymbol{x}_n, \boldsymbol{y}_n)\}_{n=1}^{N_{\text{IN}}}$ to denote the input dataset of the following sufficiency tree search algorithm and necessity tree search algorithm, and use $\mathcal{D}_{\text{OUT}}$ to denote the output set. Both the sufficiency tree and the necessity tree grow *from bottom to top*. Let $K$ denote the depth of these trees (i.e., the number of iterations), and we use $k = 1, \ldots, K$ to denote the $k$-th iteration to produce the $(k+1)$-th layer of the tree.

### C.1  MINING SUFFICIENT SET WITH TREE SEARCH

We begin by introducing our assumption on the transitivity of sufficiency. We assume that sufficiency is transitive among sets. Formally, given arbitrary three set, denoted as $\mathcal{D}_{\text{A}}, \mathcal{D}_{\text{B}}, \mathcal{D}_{\text{C}}$, if $\mathcal{D}_{\text{A}}$ is a sufficient set of $\mathcal{D}_{\text{B}}$ and $\mathcal{D}_{\text{B}}$ is a sufficient set of $\mathcal{D}_{\text{C}}$, then we can derive that $\mathcal{D}_{\text{A}}$ is a sufficient set of $\mathcal{D}_{\text{C}}$. The intuition behind the above assumption is that LLMs are proven to hold a chain-of-thought reasoning ability (Wei et al., 2022).

We leverage the transitivity of sufficiency to develop a sufficiency tree search algorithm to efficiently mine a sufficient set from the given dataset $\mathcal{D}_{\text{IN}}$. We summarize the process of building the sufficiency tree and generate the sufficiency set in Algorithm 1, where the key operations lie in lines 1, 1, and 1 to use the sufficient one to replace each pair of nodes. We notice that the output of the algorithm often includes more than just one possibility, since the sufficient set of the given dataset is usually not only one.

From the algorithm, we can see that we generate the tree from the bottom (including $|\mathcal{D}_{\text{IN}}|$ nodes) to the top (including one root node) by one-to-one comparisons shown in lines 1 and 1. Thus, the depth of the tree is $O(\log_2^{|\mathcal{D}_{\text{IN}}|})$, and the widths of layers in the tree are $|\mathcal{D}_{\text{IN}}|, |\mathcal{D}_{\text{IN}}|/2, \ldots, 1$.

### C.2  MINING NECESSARY PLUG-IN SET WITH TREE SEARCH

The intuition behind building the necessity tree is that if unplugging $\mathcal{D}_{\text{A}}$ could cause the outputs to at least one input in $\mathcal{D}_{\text{C}}$ from correct to incorrect, then unplugging $\mathcal{D}_{\text{A}} \cup \mathcal{D}_{\text{B}}$ also can not maintain the outputs to all the input in $\mathcal{D}_{\text{C}}$ correct. In other words, necessity is transitive through the inclusion

---

**Algorithm 2:** Necessity Tree Search Algorithm in FEEDER

---

**Input:** Input dataset $\mathcal{D}_{\mathtt{IN}} = \{(\boldsymbol{x}_n, \boldsymbol{y}_n)\}_{n=1}^{N_{\mathtt{IN}}}$.
**Output:** A necessary plug-in set of $\mathcal{D}_{\mathtt{IN}}$, denoted as $\mathcal{D}_{\mathtt{OUT}}$.
Initialize $k = 1$ and $\mathscr{H}_0 = \emptyset$.
**foreach** *instance* $(\boldsymbol{x}_n, \boldsymbol{y}_n)$ *in* $\mathcal{D}_{\mathtt{IN}}$ **do**
  Check $Y_{(\{\boldsymbol{x}_{n'}|\boldsymbol{x}_{n'} \in \mathcal{D}_{\mathtt{IN}}\})} = \mathbf{1}_{N_{\mathtt{IN}}}|\mathtt{unplug}((\boldsymbol{x}_n, \boldsymbol{y}_n)); C = \mathcal{D}_{\mathtt{IN}}$ (a).
  If (a) holds, let $\mathcal{H}_n = \{(\boldsymbol{x}_n, \boldsymbol{y}_n)\}$ and append $\mathcal{H}_n$ to $\mathscr{H}_0$.
**end**
**repeat**
Initialize $\mathscr{H}_k = \emptyset$.
**foreach** *pair* $(\mathcal{H}_i, \mathcal{H}_j)$ *where* $\mathcal{H}_i, \mathcal{H}_j \in \mathscr{H}_{k-1}$ **do**
  Check $Y_{(\{\boldsymbol{x}_n|\boldsymbol{x}_n \in \mathcal{D}_{\mathtt{IN}}\})} = \mathbf{1}_{N_{\mathtt{IN}}}|\mathtt{unplug}(\mathcal{H}_i \cup \mathcal{H}_j); C = \mathcal{D}_{\mathtt{IN}}$ (b).
  If (b) holds, generate a new node $\mathcal{H}_i \cup \mathcal{H}_j$, append it to $\mathscr{H}_k$, and assign $\mathcal{H}_i \cup \mathcal{H}_j$ with
   MAINTAIN signals; otherwise, append $\mathcal{H}_i$ and $\mathcal{H}_j$ to $\mathscr{H}_k$.
**end**
Assign $\mathcal{H}_{\mathtt{MAX}} = \arg\max_{\mathcal{H}_. \in \mathscr{H}_k} |\mathcal{H}_.|$ with MAINTAIN signal.
Remove the nodes without MAINTAIN signals in $\mathscr{H}_k$.
Grow tree from bottom to top via $k = k + 1$.
**until** $|\mathscr{H}_k| = 1$ where we assume the iteration is $K$.
Let $\mathcal{H}_{\mathtt{UNNCESSARY}}$ denote only one element (i.e. the root node) in $\mathscr{H}_K$.
Assign $\mathcal{D}_{\mathtt{OUT}}$ as removing $\mathcal{H}_{\mathtt{UNNCESSARY}}$ from $\mathcal{D}_{\mathtt{IN}}$, i.e., $\mathcal{D}_{\mathtt{OUT}} = \mathcal{D}_{\mathtt{IN}} - \mathcal{H}_{\mathtt{UNNECESSARY}}$.

---

relation of sets, namely if unplugging a subset would degrade the performance, then unplugging the whole set would also degrade the performance.

We summarize the process of building the necessity tree and generate the necessary set in Algorithm 2, where MAINTAIN signals are introduced as a pruning technique, as we only need to take care of those nodes whose size would increase (i.e., assigning MAINTAIN signals to newly generated nodes in line 2), and the nodes with the maximum size (i.e., assigning a MAINTAIN signal to the node with the maximum size in line 2). From the algorithm, we can see that the computation of each iteration is super costly, as it requires running checker for $O(\mathtt{C}_{N_{\mathtt{IN}}}^2)$ times (where $\mathtt{C}_.$ is a combination operator), which is infeasible in many real-world scenarios.

Therefore, we propose an alternative algorithm as summarized in Algorithm 3, where we introduce the maximum number of rounds $R_{\mathtt{MAX}}$ to trade-off the effectiveness and efficiency. From the algorithm, we can see that for each round, we need to build a tree whose depth is $O(\log_2^{N_{\mathtt{IN}}})$ (which is similar to Algorithm 1). Comparing to Algorithm 2, Algorithm 3 introduces $R_{\mathtt{MAX}}$ to control the complexity. If we want to obtain a conceptually necessary plug-in set of the given dataset, then we should assign $R_{\mathtt{MAX}} = +\infty$.

In the following, we prove that by combining our sufficiency tree search algorithm (i.e., Algorithm 1) and necessity tree search algorithm (i.e., either Algorithm 2 or Algorithm 3), the resulting set is a parsimonious set of $\mathcal{D}_{\mathtt{TRAIN}}$. Formally, we have the following theorem.

**Theorem 3.** *If we successively apply the sufficiency tree search algorithm and the necessity tree search algorithm (either the original one or the alternative one) on $\mathcal{D}_{\mathtt{TRAIN}}$ to obtain $\widetilde{\mathcal{D}}_{\mathtt{TRAIN}}$, then $\widetilde{\mathcal{D}}_{\mathtt{TRAIN}}$ is a parsimonious set of $\mathcal{D}_{\mathtt{TRAIN}}$.*

*Proof.* According to Theorem 2, to prove the above theorem, we only need to prove that $\widetilde{\mathcal{D}}_{\mathtt{TRAIN}}$ is a sufficient set of $\mathcal{D}_{\mathtt{TRAIN}}$ and a necessary set of $\mathcal{D}_{\mathtt{TRAIN}}$. For convenience, we use $\widehat{\mathcal{D}}_{\mathtt{TRAIN}}$ to denote the output of the sufficient tree search algorithm (i.e., Algorithm 1).

We begin by proving sufficiency. Firstly, for each data point in $\mathcal{D}_{\mathtt{TRAIN}}$, the root node in the tree is a sufficient set for each leaf node in the tree. Formally, we have $Y_{\{\boldsymbol{x}_n|\boldsymbol{x}_n \in \mathcal{D}_{\mathtt{TRAIN}}\}} = \mathbf{1}_N|\mathtt{plug}(\widehat{\mathcal{D}}_{\mathtt{TRAIN}}), Z = \emptyset$, meaning that the resulting set $\widehat{\mathcal{D}}_{\mathtt{TRAIN}}$ is a sufficient set of $\mathcal{D}_{\mathtt{TRAIN}}$.

Then, either Algorithm 2 or 3 preserves the sufficiency during the necessity checker, i.e., lines 2 and 2 in Algorithm 2, and lines 3 and 3 in Algorithm 3. In other words, we have:

$$Y_{(\{\boldsymbol{x}_n|\boldsymbol{x}_n \in \widehat{\mathcal{D}}_{\mathtt{TRAIN}}\})} = \mathbf{1}_{\widehat{N}}|\mathtt{unplug}(\widehat{\mathcal{D}}_{\mathtt{TRAIN}} - \widetilde{\mathcal{D}}_{\mathtt{TRAIN}}); C = \widehat{\mathcal{D}}_{\mathtt{TRAIN}}, \tag{23}$$

---

**Algorithm 3:** Alternative Necessity Tree Search Algorithm in `FEEDER`

---

**Input:** Input dataset $\mathcal{D}_{\text{IN}} = \{(\boldsymbol{x}_n, \boldsymbol{y}_n)\}_{n=1}^{N_{\text{IN}}}$, maximum number of rounds $R_{\text{MAX}}$.
**Output:** A necessary plug-in set of $\mathcal{D}_{\text{IN}}$, denoted as $\mathcal{D}_{\text{OUT}}$.
Initialize the number of rounds $r = 0$, and the set of unnecessary data $\mathcal{D}_r = \emptyset$.
**repeat**
Initialize $k = 1$ and $\mathscr{H}_0 = \emptyset$.
Update input dataset by removing the unnecessary part $\mathcal{D}_{\text{IN}} = \mathcal{D}_{\text{IN}} - \mathcal{D}_r$.
**foreach** *instance* $(\boldsymbol{x}_n, \boldsymbol{y}_n)$ *in* $\mathcal{D}_{\text{IN}}$ **do**
    Check $Y_{(\{\boldsymbol{x}_{n'}|\boldsymbol{x}_{n'} \in \mathcal{D}_{\text{IN}}\})} = \mathbf{1}_{N_{\text{IN}}} | \texttt{unplug}((\boldsymbol{x}_n, \boldsymbol{y}_n)); C = \mathcal{D}_{\text{IN}}$ (a). If (a) holds, let
    $\mathcal{H}_n = \{(\boldsymbol{x}_n, \boldsymbol{y}_n)\}$ and append $\mathcal{H}_n$ to $\mathscr{H}_0$.
**end**
**repeat**
Initialize $\mathscr{H}_k = \emptyset$.
**foreach** *pair* $(\mathcal{H}_i, \mathcal{H}_j)$ *where* $\mathcal{H}_i, \mathcal{H}_j \in \mathscr{H}_{k-1}$ **do**
    Check $Y_{(\{\boldsymbol{x}_n|\boldsymbol{x}_n \in \mathcal{D}_{\text{IN}}\})} = \mathbf{1}_{N_{\text{IN}}} | \texttt{unplug}(\mathcal{H}_i \cup \mathcal{H}_j); C = \mathcal{D}_{\text{IN}}$ (b).
    If (b) holds, generate a new node $\mathcal{H}_i \cup \mathcal{H}_j$, append it to $\mathscr{H}_k$; otherwise, append $\mathcal{H}_i$ and $\mathcal{H}_j$ to
    $\mathscr{H}_k$.
    Remove $\mathcal{H}_i, \mathcal{H}_j$ from $\mathscr{H}_{k-1}$, i.e., $\mathscr{H}_{k-1} - \{\mathcal{H}_i, \mathcal{H}_j\}$.
**end**
Grow tree from bottom to top via $k = k + 1$.
**until** $|\mathscr{H}_k| = 1$ where we assume the iteration is $K$.
Let $\mathcal{H}_{\text{UNNCESSARY}}$ denote only one element (i.e. the root node) in $\mathscr{H}_K$.
Update the number of rounds, i.e., $r = r + 1$.
Update $\mathcal{D}_r$ to include the unnecessary part $\mathcal{H}_{\text{UNNCESSARY}}$, i.e., $\mathcal{D}_r = \mathcal{D}_r \cup \mathcal{H}_{\text{UNNCESSARY}}$.
**until** $|\mathcal{H}_{\text{UNNCESSARY}}| \leq 1$ or $r \geq R_{\text{MAX}}$.
Assign $\mathcal{D}_{\text{OUT}}$ as removing $\mathcal{D}_r$ from $\mathcal{D}_{\text{IN}}$, i.e., $\mathcal{D}_{\text{OUT}} = \mathcal{D}_{\text{IN}} - \mathcal{D}_r$.

---

where $\widehat{N} = |\widehat{\mathcal{D}}_{\text{TRAIN}}|$. It can be re-written as $Y_{(\{\boldsymbol{x}_n|\boldsymbol{x}_n \in \widehat{\mathcal{D}}_{\text{TRAIN}}\})} = \mathbf{1}_{\widehat{N}} | \texttt{plug}(\widetilde{\mathcal{D}}_{\text{TRAIN}}); C = \emptyset$, showing that plugging-in $\widetilde{\mathcal{D}}_{\text{TRAIN}}$ is sufficient for $\widehat{\mathcal{D}}_{\text{TRAIN}}$.

Combining $\widehat{\mathcal{D}}_{\text{TRAIN}}$ being a sufficient set of $\mathcal{D}_{\text{TRAIN}}$ and $\widetilde{\mathcal{D}}_{\text{TRAIN}}$ being a sufficient set of $\widehat{\mathcal{D}}_{\text{TRAIN}}$, we arrive at $\widetilde{\mathcal{D}}_{\text{TRAIN}}$ is a sufficient set of $\mathcal{D}_{\text{TRAIN}}$.

Next, we investigate necessity. Our goal is to prove unplugging any data point in $\widetilde{\mathcal{D}}_{\text{TRAIN}}$ would lead to a degradation of the LLM's performance. For convenience, we use $(\boldsymbol{x}_n, \boldsymbol{y}_n) \in \widetilde{\mathcal{D}}_{\text{TRAIN}}$ to denote an arbitrary data point. If we applying Algorithm 2 to execute the necessary tree search algorithm, then $(\boldsymbol{x}_n, \boldsymbol{y}_n)$ must be in $\mathscr{H}_0$, or out of $\mathscr{H}_0$.

If $(\boldsymbol{x}_n, \boldsymbol{y}_n)$ is not an element in $\mathscr{H}_0$. Then, according to the computing process of $\mathscr{H}_0$ (shown in lines 1 and 1 in Algorithm 2), unplugging $(\boldsymbol{x}_n, \boldsymbol{y}_n)$ it would definitely cause the LLM's performance on the input dataset $\widehat{\mathcal{D}}_{\text{TRAIN}}$ from $Y_{(\{\boldsymbol{x}_n|\boldsymbol{x}_n \in \widehat{\mathcal{D}}_{\text{TRAIN}}\})} = \mathbf{1}_{\widehat{N}}$ to $Y_{(\{\boldsymbol{x}_n|\boldsymbol{x}_n \in \widehat{\mathcal{D}}_{\text{TRAIN}}\})} \neq \mathbf{1}_{\widehat{N}}$.

If $(\boldsymbol{x}_n, \boldsymbol{y}_n)$ is an element in $\mathscr{H}_0$, then according to lines 1, 1 and 1 in Algorithm 2, $(\boldsymbol{x}_n, \boldsymbol{y}_n)$ must be in $\mathcal{H}_{\text{UNNECESSARY}}$; otherwise, $\mathcal{H}_{\text{UNNECESSARY}} \cup \{(\boldsymbol{x}_n, \boldsymbol{y}_n)\}$ should be $\mathcal{H}_{\text{MAX}}$ and always stay in $\mathcal{H}$. until becoming the root node (i.e., $\mathcal{H}_{\text{UNNECESSARY}}$ should be updated to be $\mathcal{H}_{\text{UNNECESSARY}} \cup \{(\boldsymbol{x}_n, \boldsymbol{y}_n)\}$). Thus, $(\boldsymbol{x}_n, \boldsymbol{y}_n)$ must be in $\mathcal{H}_{\text{UNNECESSARY}}$. However, all the data points in $\mathcal{H}_{\text{UNNECESSARY}}$ are removed from $\widehat{\mathcal{D}}_{\text{TRAIN}}$, causing a contradiction. Hence, unplugging $(\boldsymbol{x}_n, \boldsymbol{y}_n)$ would change the LLM's performance, namely necessity holds.

Then, we consider the case where we apply Algorithm 3 to execute the necessary tree search.

Similarly, if $(\boldsymbol{x}_n, \boldsymbol{y}_n)$ is not selected by the necessity checker shown in line 3 in Algorithm 3, then unplugging $(\boldsymbol{x}_n, \boldsymbol{y}_n)$ would definitely cause a degradation of the LLM's performance.

If $(\boldsymbol{x}_n, \boldsymbol{y}_n)$ is selected by the necessity checker, then $(\boldsymbol{x}_n, \boldsymbol{y}_n)$ must be included in $\mathcal{D}_r$; otherwise, $\mathcal{D}_r$ would continue to update, since the condition of stopping iteration is that there is no or only one unnecessary node. However, all the data points are removed from $\widehat{\mathcal{D}}_{\text{TRAIN}}$, causing a contradiction. Hence, unplugging $(\boldsymbol{x}_n, \boldsymbol{y}_n)$ would change the LLM's performance, namely necessity holds.

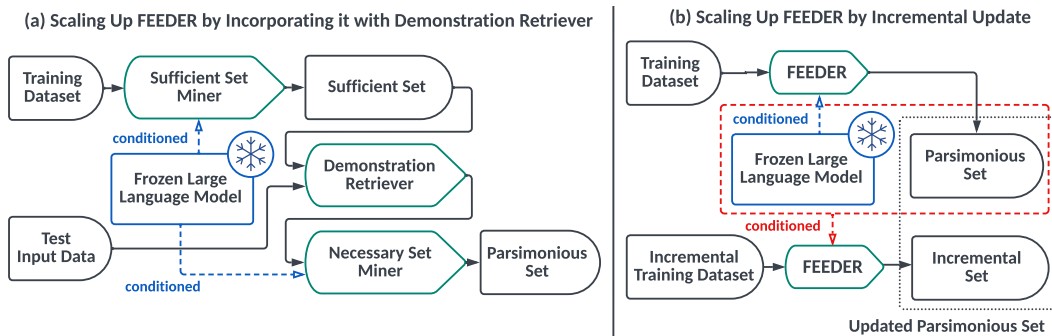

Figure 5: Scaling up FEEDER to real-world applications, where (a) we first run the sufficiency tree algorithm, then run a retriever to retrieve items according to test data, and run the necessity tree algorithm to remove the unnecessary part from the retrieved data; (b) we develop an incremental update algorithm to avoid re-computing over all the unchanged and changed training examples, which can both used to re-compute the sufficiency and necessary parsimonious set of the given datasets, and used to incrementally select each demonstration.

Combining the above sufficiency and necessity, we can conclude that $\widetilde{\mathcal{D}}_{\texttt{TRAIN}}$ is a parsimonious data for $\mathcal{D}_{\texttt{TRAIN}}$. $\qquad\square$

# D    SCALING UP FEEDER INTO REAL-WORLD APPLICATIONS

## D.1    FITTING FEEDER TO REAL-WORLD CASES

We summarize two main limitations of FEEDER in real-world applications.

**Scaling Up FEEDER by incorporating it with Demonstration Retriever.** One limitation is that the produced $\mathcal{D}_{\texttt{FEED}}$ is too large to be directly used as input demonstrations. For this purpose, we incorporate FEEDER to existing demonstration retrievers to retrieve relevant demonstrations from $\mathcal{D}_{\texttt{FEED}}$, namely first mining a parsimonious set, and then retrieval. In this regard, our approach can be used as a core-set selection method that mines informative training samples from the whole training dataset, to benefit the downstream tasks. However, directly applying the necessity search algorithm on sufficiency-filtered data could also be problematic, since sufficiency-filtered demonstrations would be too large to be prompted. Therefore, we design to place the retriever between the sufficient plug-in data miner and the necessary plug-in data miner. Then, we only need to filter the unnecessary parts from the retrieved demonstrations. We depict the above resulting process in Figure 5(a). Here, the input is the whole training dataset, and the output is the parsimonious set, and thus the whole process corresponds to FEEDER component in Figure 1. We note that we could run the sufficiency tree search *multiple rounds* to obtain a condensed sufficient set before employing the retriever.

**Scaling Up FEEDER by Incremental Update.** Another limitation is that many real-world (training) datasets are temporal (and some even require daily updates), and directly re-calculating the parsimonious set over all the unchanged and changed samples is time-consuming. To address this, we develop a new incremental update algorithm for FEEDER allowing us to only re-compute the changed parts (including new adding and modified samples). As shown in Figure 5(b), once we generate the golden parsimonious set for the original dataset, then we can treat the unchanged part of plug-and-play plug-in data and the LLM as the whole (shown as the dashed box), as a new "LLM model", and therefore, we only apply FEEDER to compute incremental data for the changed part (including newly added and modified data points). When incorporating into the above process shown in Figure 5(a), we could store and update the sufficient plug-in database instead of the parsimonious database, and treat the retrieved data, instead of the whole plug-in data, and the LLM as a whole.

## D.2    IMPLEMENTATION DETAILS

For each dataset, we directly follow the official splits to obtain $\mathcal{D}_{\texttt{TRAIN}}$ and $\mathcal{D}_{\texttt{TEST}}$. We further introduce detailed descriptions of three different retrievers in our paper. The first one is a random retriever, denoted as RAN, which randomly retrieves samples from the retrieval pool. The second one is a similarity based retriever, denoted as SIM, which retrieves samples similar to the test samples. Formally, let $\mathcal{D}_{\texttt{RETRIEVE}}$ denote the retrieval pool. Then, for each test sample $\boldsymbol{x}_m$, the metric of SIM can be written as:

$$\texttt{SIM}(\boldsymbol{x}_m, \boldsymbol{x}_n) = \texttt{COS}(\texttt{TRANSFORMER}(\boldsymbol{x}_m), \texttt{TRANSFORMER}(\boldsymbol{x}_n)), \text{ where } \boldsymbol{x}_n \in \mathcal{D}_{\texttt{RETRIEVE}}, \quad (24)$$

Table 4: Performance comparisons on text classification datasets for the in-context learning setting. We report both the mean and variance of accuracy using four different seeds and four different permutations of n-shots.

| $\Psi_{\text{LLM}}(\cdot)$ | $\widetilde{\mathcal{D}}_{\text{TRAIN}}$ | $n$ | FPB | | | SST-2 | | | COLA | | |
|---|---|---|---|---|---|---|---|---|---|---|---|
| | | | RAN | SIM | DIV | RAN | SIM | DIV | RAN | SIM | DIV |
| MED | $\mathcal{D}_{\text{TRAIN}}$ | 1 | 27.2 (6.1) | 25.3 (0.1) | 25.3 (0.1) | 48.9 (4.6) | 24.5 (0.2) | 24.5 (0.2) | 29.0 (5.4) | 38.8 (0.1) | 38.8 (0.1) |
| | | 2 | 27.4 (6.2) | 45.8 (0.2) | 40.4 (0.1) | 51.2 (5.8) | 62.5 (0.2) | 62.5 (0.2) | 30.9 (4.6) | 38.5 (0.2) | 36.2 (0.1) |
| | | 5 | 26.3 (4.5) | 55.9 (0.1) | 44.7 (0.2) | 62.6 (5.6) | 79.4 (0.2) | 61.7 (0.1) | 69.4 (5.8) | 49.3 (0.1) | 47.0 (0.2) |
| | | 10 | 27.8 (5.1) | 63.1 (0.1) | 50.7 (0.1) | 50.9 (4.9) | 83.8 (0.3) | 76.9 (0.2) | 31.6 (4.6) | 52.5 (0.2) | 58.8 (0.2) |
| | $\mathcal{D}_{\text{FEED}}$ | 1 | **28.4** (3.4) | **28.8** (2.1) | **28.8** (2.1) | **49.8** (4.2) | **48.1** (1.9) | **48.1** (1.9) | 29.4 (4.6) | 35.1 (1.5) | 35.1 (1.5) |
| | | 2 | **35.5** (4.3) | 47.4 (2.6) | 37.9 (1.9) | **67.3** (4.4) | 67.7 (1.4) | **64.7** (1.5) | 31.1 (2.2) | **41.7** (1.2) | 34.9 (1.9) |
| | | 5 | **28.3** (3.0) | 54.6 (1.7) | **47.9** (1.0) | **70.3** (4.4) | 77.9 (1.2) | **68.5** (1.9) | 65.2 (2.0) | **57.3** (1.2) | **54.6** (1.7) |
| | | 10 | **39.6** (3.4) | 63.2 (2.3) | 49.8 (1.2) | **75.2** (6.2) | 83.0 (1.7) | 77.2 (1.5) | **69.3** (3.8) | **68.7** (2.4) | **68.5** (2.9) |
| LAR | $\mathcal{D}_{\text{TRAIN}}$ | 1 | 33.8 (5.2) | 29.9 (0.1) | 29.9 (0.1) | 49.0 (4.3) | 42.3 (0.2) | 42.3 (0.2) | 22.1 (5.7) | 38.3 (0.1) | 38.3 (0.1) |
| | | 2 | 27.0 (6.1) | 55.4 (0.2) | 49.9 (0.3) | 68.0 (5.2) | 70.7 (0.1) | 59.6 (0.2) | 41.1 (4.2) | 36.8 (0.2) | 37.7 (0.1) |
| | | 5 | 27.2 (4.8) | 64.3 (0.1) | 45.1 (0.3) | 49.1 (4.3) | 80.6 (0.1) | 67.5 (0.2) | 66.2 (4.7) | 53.8 (0.2) | 48.5 (0.3) |
| | | 10 | 47.0 (5.5) | 65.5 (0.2) | 52.9 (0.1) | 71.1 (4.5) | 84.6 (0.1) | 73.1 (0.2) | 43.4 (4.5) | 55.5 (0.2) | 56.1 (0.4) |
| | $\mathcal{D}_{\text{FEED}}$ | 1 | 33.8 (4.4) | **32.6** (0.7) | **32.6** (0.7) | 49.1 (3.0) | **47.7** (1.3) | **47.7** (1.3) | **29.6** (3.8) | 35.1 (1.1) | 35.1 (1.1) |
| | | 2 | **37.5** (4.2) | 54.8 (1.1) | 47.6 (1.3) | 67.8 (3.8) | **73.0** (2.9) | 61.2 (2.1) | 36.6 (3.5) | 37.0 (2.8) | **34.6** (2.0) |
| | | 5 | **38.9** (3.3) | 64.5 (1.3) | **48.0** (2.7) | 59.3 (2.4) | 80.9 (1.3) | **69.6** (1.7) | 69.2 (3.3) | 68.6 (1.6) | **66.6** (1.7) |
| | | 10 | **63.5** (2.8) | 64.7 (1.6) | **50.1** (1.5) | **76.0** (3.0) | 84.7 (1.4) | **75.6** (1.8) | 69.3 (4.8) | 68.8 (2.0) | **68.9** (1.8) |
| NEO | $\mathcal{D}_{\text{TRAIN}}$ | 1 | 54.9 (3.9) | 61.6 (0.1) | 61.6 (0.1) | 49.2 (3.7) | 33.8 (0.1) | 33.8 (0.1) | 25.5 (3.4) | 36.5 (0.2) | 36.5 (0.2) |
| | | 2 | 53.6 (4.0) | 66.8 (0.2) | 60.0 (0.1) | 76.8 (3.5) | 81.5 (0.1) | 76.3 (0.4) | 30.7 (3.1) | 55.5 (0.2) | 56.5 (0.4) |
| | | 5 | 28.2 (4.0) | 68.2 (0.1) | 60.4 (0.1) | 65.1 (3.5) | 66.1 (0.3) | 55.9 (0.1) | 40.0 (3.6) | 55.9 (0.1) | 52.5 (0.2) |
| | | 10 | 49.0 (4.8) | 75.8 (0.1) | 71.1 (0.2) | 69.8 (4.8) | 84.1 (0.1) | 69.7 (0.1) | 69.6 (4.5) | 59.3 (0.3) | 63.4 (0.1) |
| | $\mathcal{D}_{\text{FEED}}$ | 1 | **58.1** (4.7) | 61.8 (1.4) | 61.8 (1.4) | 49.3 (5.1) | **48.3** (1.9) | **48.3** (1.9) | **28.3** (5.4) | 34.8 (1.3) | 34.8 (1.3) |
| | | 2 | **61.4** (3.3) | 64.1 (1.5) | 58.8 (1.1) | 75.1 (2.8) | **82.6** (2.1) | **78.5** (1.9) | **69.3** (3.7) | **64.7** (1.4) | **64.7** (1.6) |
| | | 5 | **43.2** (2.6) | 68.8 (1.8) | **62.7** (1.3) | 73.2 (4.2) | **82.9** (2.7) | **71.6** (2.4) | **68.7** (3.2) | 67.2 (2.4) | **65.8** (1.8) |
| | | 10 | **61.4** (2.3) | 74.8 (1.9) | 71.9 (1.8) | 72.4 (3.4) | **85.8** (2.5) | **71.8** (2.9) | 69.8 (2.8) | **68.8** (1.4) | **68.9** (1.3) |
| LLA | $\mathcal{D}_{\text{TRAIN}}$ | 1 | 29.0 (4.7) | 47.1 (0.1) | 47.1 (0.1) | 48.2 (2.9) | 47.0 (0.1) | 46.2 (0.1) | 38.9 (6.7) | 41.2 (0.2) | 41.2 (0.2) |
| | | 2 | 27.4 (3.4) | 68.4 (0.2) | 67.1 (0.3) | 67.8 (3.2) | 68.7 (0.2) | 67.5 (0.1) | 43.5 (4.5) | 47.4 (0.2) | 49.6 (0.1) |
| | | 5 | 39.7 (3.2) | 80.3 (0.2) | 78.9 (0.1) | 75.2 (3.3) | 80.7 (0.1) | 77.8 (0.2) | 50.2 (3.7) | 52.6 (0.2) | 48.2 (0.3) |
| | | 10 | 37.9 (2.6) | 87.4 (0.3) | 86.5 (0.2) | 82.1 (3.8) | 87.6 (0.1) | 86.5 (0.2) | 59.6 (4.3) | 55.3 (0.2) | 60.0 (0.4) |
| | $\mathcal{D}_{\text{FEED}}$ | 1 | **33.7** (5.3) | **51.7** (0.8) | **51.7** (0.8) | **49.6** (2.4) | **51.3** (1.6) | **51.3** (1.6) | **41.2** (2.1) | **43.8** (1.8) | **43.8** (1.8) |
| | | 2 | **39.6** (5.0) | 68.7 (1.5) | **69.8** (0.7) | 63.5 (2.5) | 65.7 (4.2) | 66.1 (2.1) | **50.8** (2.3) | **48.6** (1.7) | 43.5 (1.3) |
| | | 5 | **45.6** (4.8) | **87.9** (4.8) | 77.5 (3.4) | **77.6** (4.0) | 81.0 (1.3) | **79.4** (1.0) | **53.8** (2.8) | **55.3** (1.6) | **51.8** (1.4) |
| | | 10 | 37.8 (6.4) | 87.1 (3.9) | **87.8** (2.2) | **83.8** (2.8) | 86.4 (2.0) | 87.2 (1.3) | **59.5** (3.1) | **64.0** (1.9) | **65.4** (2.0) |

where $\texttt{COS}(\cdot)$ denotes a cosine similarity metric, and $\texttt{TRANSFORMER}(\cdot)$ denotes a sentence transformer (Reimers & Gurevych, 2019). Here, we directly use the Sentence Transformers library[1] from Hugging Face in our implementation. Then, we are able to retrieve $N_{\texttt{shot}}$ samples with maximum $\texttt{SIM}$ values from $\mathcal{D}_{\texttt{RETRIEVE}}$. The third one is a diversity based retriever, denoted as $\texttt{DIV}$, where we adopt the maximal marginal relevance method (Carbonell & Goldstein, 1998) as the metric of $\texttt{DIV}$.

Formally, we have:

$$\texttt{DIV}(\boldsymbol{x}_m, \boldsymbol{x}_n) = \texttt{SIM}(\boldsymbol{x}_m, \boldsymbol{x}_n) - \eta \cdot \max_{\boldsymbol{x}_{n'} \in \mathcal{D}'_{\texttt{RETRIEVE}}} \texttt{SIM}(\boldsymbol{x}_m, \boldsymbol{x}_{n'}), \text{ where } \boldsymbol{x}_n \in \mathcal{D}_{\texttt{RETRIEVE}} - \mathcal{D}'_{\texttt{RETRIEVE}},$$

(25)

where $\mathcal{D}'_{\texttt{RETRIEVE}}$ denotes the set of previously selected instances. We can see that $\texttt{DIV}$ prefers the instance that is both similar to the test samples meanwhile distant to previously selected instances. $\eta$ is a hyper-parameter to balance the above two parts. We set $\eta = 1$ in our experiment.

We list some key hyper-parameters used in our supervised learning setting: the batch size is set as 32, the warm steps is set as 100, the learning rate is set as $5 \times 10^{-5}$, and the weight decay is set as 0.01.

### D.3 Additional Results for In-Context Learning

We report performance comparison results on text classification datasets SUBJ, SST-5, and TREC datasets in Table 1. We include the results of COLA, SST-2, and FPB datasets in Table 4.

To further evaluate $\texttt{FEEDER}$ on reasoning dataset GSM8K (Cobbe et al., 2021) and semantic-parsing dataset SMCALFlow (Andreas et al., 2020) with one GPT-3 variant with 6B parameters as the LLM. We use $\texttt{SIX}$ to denote the LLM base.

We summarize the results in Table 5. The table shows that our $\texttt{FEEDER}$ exhibits adaptability to more intricate tasks and consistently enhances the performance of the LLM. Furthermore, it is evident that tasks involving reasoning and semantic parsing pose significant challenges for the 6B-sized LLM, making it difficult to achieve satisfactory performance.

---

[1] https://huggingface.co/sentence-transformers

Table 5: Performance comparisons on reasoning GSM8K dataset and semantic-parsing SMCALFlow dataset for gpt-j-6B LLM base with different retrievers RAN, SIM) in 1-shot and 2-shot settings. We report both the mean and variance of accuracy using four different seeds and four different permutations of n-shots.

| $\Psi_{\text{LLM}}(\cdot)$ | $\widetilde{\mathcal{D}}_{\text{TRAIN}}$ | $n$ | GSM8K | | SMCALFlow | |
|---|---|---|---|---|---|---|
| | | | RAN | SIM | RAN | SIM |
| SIX | $\mathcal{D}_{\text{TRAIN}}$ | 1 | 1.21 (0.83) | 2.81 (0.12) | 1.78 (0.72) | 8.95 (0.19) |
| | | 2 | 1.44 (0.65) | 4.01 (0.13) | 2.67 (0.98) | 9.91 (0.20) |
| | $\mathcal{D}_{\text{FEED}}$ | 1 | **2.27** (0.49) | **3.03** (0.15) | **2.35** (0.59) | **9.19** (0.08) |
| | | 2 | **2.80** (0.53) | **4.16** (0.14) | **3.51** (0.71) | **10.73** (0.07) |

Table 6: Performance comparisons on text classification COLA dataset for different LLM bases with different retrievers IP and ET. We report both the mean and variance of accuracy using four different seeds and four different permutations of n-shots.

| $\widetilde{\mathcal{D}}_{\text{TRAIN}}$ | $n$ | MED | | LAR | | NEO | |
|---|---|---|---|---|---|---|---|
| | | IP | ET | IP | ET | IP | ET |
| $\mathcal{D}_{\text{TRAIN}}$ | 1 | 37.8 (0.3) | 37.6 (0.4) | 43.5 (0.4) | 42.5 (0.3) | 26.7 (0.2) | 27.4 (0.3) |
| | 2 | 39.2 (0.2) | 39.4 (0.5) | 41.4 (0.6) | 41.6 (0.4) | 28.5 (0.4) | 31.0 (0.3) |
| | 5 | 53.5 (0.5) | 52.8 (0.3) | 66.9 (0.7) | 65.4 (0.2) | 41.3 (0.4) | 40.8 (0.1) |
| | 10 | 54.1 (0.6) | 52.9 (0.4) | 42.5 (0.5) | 46.7 (0.2) | 71.6 (0.4) | 70.4 (0.2) |
| $\mathcal{D}_{\text{FEED}}$ | 1 | 35.9 (0.3) | 34.7 (0.3) | **48.6** (0.5) | **48.4** (0.3) | 26.6 (0.3) | 27.6 (0.5) |
| | 2 | **43.5** (0.5) | **42.7** (0.1) | 40.5 (0.7) | 41.4 (0.6) | **69.9** (0.3) | **67.4** (0.2) |
| | 5 | 54.5 (0.3) | **55.0** (0.4) | **69.6** (0.5) | **68.8** (0.2) | **69.7** (0.2) | **68.7** (0.1) |
| | 10 | **69.4** (0.3) | **67.8** (0.4) | **69.9** (0.1) | **69.4** (0.7) | 72.4 (0.2) | **72.7** (0.2) |

Besides three basic demonstration retrievers introduced in Section 5, we also examine the performance of FEEDER with some active learning techniques as the retrievers. One (Köksal et al., 2022), denoted as IP, uses inter-prompt uncertainty sampling to get demonstrations. The other (Roy & McCallum, 2001), denoted as ET, adopts entropy-based sampling techniques to retrieve demonstrations. We summarize the corresponding results in Table 6. These results verify that our FEEDER can collaborate with various demonstration retrievers.

## D.4 ADDITIONAL RESULTS FOR FINE-TUNING

We report performance comparison results on text classification datasets SUBJ, SST-5, and TREC datasets in Table 2. We include the results of COLA, SST-2, and FPB datasets in Table 7.

## D.5 COMPLEXITY STUDY

We report the time complexity of running the sufficient tree search algorithm (i.e., Algorithm 1) on COLA and TREC datasets in Figure 6. From the figure, we can observe that as the number of samples reduces, the time consumption of Algorithm 1 would also reduce. Combining Figure 6 and 3, we can see that the time consumption is almost linear to the size of data samples.

We also test the time complexity of inference time of the LLM base NEO in the context of the in-context learning on COLA dataset in Figure 7, which includes running our necessary set miner. From the figure, we can see that our method using $\mathcal{D}_{\text{FEED}}$ instead of $\mathcal{D}_{\text{TRAIN}}$ can largely reduce the computation cost. And, since we employ Algorithm 3 with $R_{\text{MAX}} = 1$ as our necessary set miner, it almost makes no difference in the time cost.

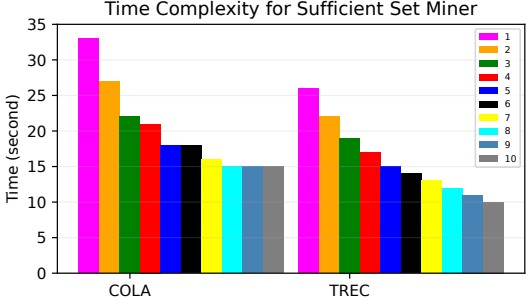

Figure 6: Time comparisons of running the sufficient tree search algorithm on COLA and TREC datasets with different rounds.

Table 7: Performance comparisons on text classification datasets for the fine-tuning setting. We report both the mean and variance of accuracy using four different seeds and four different permutations of n-shots.

| $\Psi_{\text{LLM}}(\cdot)$ | $\widetilde{\mathcal{D}}_{\text{TRAIN}}$ | $n$ | FPB | | | SST-2 | | | COLA | | |
|---|---|---|---|---|---|---|---|---|---|---|---|
| | | | RAN | SIM | DIV | RAN | SIM | DIV | RAN | SIM | DIV |
| NEO | $\mathcal{D}_{\text{TRAIN}}$ | 1 | 62.7 (5.7) | 78.4 (0.1) | 78.4 (0.1) | 41.3 (4.1) | 52.6 (0.2) | 52.8 (0.2) | 49.3 (5.2) | 68.5 (0.2) | 68.5 (0.2) |
| | | 2 | 63.1 (4.6) | 74.2 (0.3) | 73.1 (0.2) | 74.5 (3.2) | 75.8 (0.4) | 76.4 (0.5) | 70.8 (5.7) | 63.9 (0.2) | 64.3 (0.4) |
| | | 5 | 70.8 (5.1) | 73.3 (0.1) | 72.7 (0.2) | 53.6 (4.1) | 57.8 (0.3) | 57.3 (0.2) | 30.7 (4.7) | 54.4 (0.3) | 54.0 (0.3) |
| | | 10 | 62.2 (4.4) | 63.0 (0.6) | 69.6 (0.5) | 50.8 (2.9) | 55.5 (0.2) | 57.8 (0.2) | 30.7 (3.8) | 50.7 (0.3) | 47.6 (0.4) |
| | $\mathcal{D}_{\text{FEED}}$ | 1 | **73.0** (4.4) | **83.5** (1.5) | **83.5** (1.5) | **49.5** (4.1) | **56.7** (1.5) | **56.7** (1.5) | **56.8** (3.3) | **72.6** (0.9) | **72.6** (0.9) |
| | | 2 | **76.1** (3.8) | **84.1** (1.4) | **82.5** (1.7) | **75.6** (2.8) | **76.4** (0.6) | **75.2** (0.5) | **74.2** (3.7) | **71.3** (0.7) | **71.5** (0.9) |
| | | 5 | **75.7** (3.5) | **78.7** (1.5) | **83.1** (1.9) | **67.4** (2.9) | **69.5** (1.8) | **68.7** (1.9) | **71.7** (3.2) | **69.4** (2.3) | **70.0** (1.9) |
| | | 10 | **70.5** (3.3) | **75.6** (1.3) | **77.6** (1.8) | **68.9** (2.0) | **68.6** (1.6) | **69.0** (1.4) | **71.3** (2.7) | **68.5** (1.7) | **68.5** (1.9) |

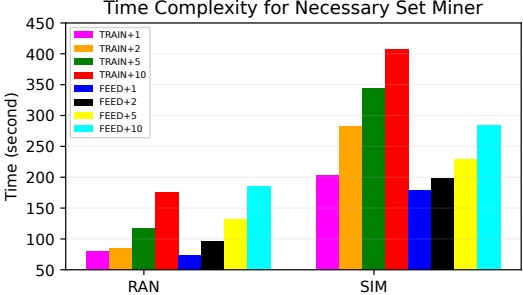

Figure 7: Time comparisons of running the LLM base NEO in the in-context learning setting (including necessary tree search algorithm) on COLA dataset with different retrievers (RAN, SIM) in different shot setting (1, 2, 5, 10). Here, we use TRAIN + $X$ to denote the result of using $\mathcal{D}_{\text{TRAIN}}$ as the retrieval pool in the setting of $X$-shot inference, and use FEED + $X$ to denote the result of using $\mathcal{D}_{\text{FEED}}$ as the retrieval pool in the setting of $X$-shot inference.

# E    CASE STUDY WITH ARTIFICIAL DATA POINTS GENERATED BY LLMS

We begin by verifying the transitivity of the sufficiency with a simple case. With gpt-3.5-turbo, we ask it with *which place does Jerry lives in?* LLM responses with *I'm sorry, but I don't have access to personal information about individuals, including your friend Jerry.* Then, let $\mathcal{D}_{\text{A}}, \mathcal{D}_{\text{B}}, \mathcal{D}_{\text{C}}$ denote the city, the country, and the continent he lives in. Then, we can observe that if we tell the LLM about the city (e.g., London), then the LLM can tell about the country (e.g., United Kingdom); if we tell the LLM about the country (e.g., United Kingdom), then the LLM can tell about the continent (e.g., Europe). Also, if we tell the LLM about the city (e.g., London), then we ask about the continent, and the LLM also can tell (e.g., Europe). This observation is one case to prove the transitivity of the sufficiency.

Then, we establish a case study to verify the key idea of this paper, i.e., whether a demonstration is high-quality or not should consider the LLM in use.

We consider the factual error made by Google Bard in the first demo[2]. Bard said *The James Webb Space Telescope took the very first pictures of a planet outside of our own solar system.* However, the fact is *The Very Large Telescope took the very first pictures of a planet outside of our own solar system.* Based on the above statements, we produce the question *What took the very first pictures of a planet outside of our own solar system?*

We ask the question to gpt-3.5-turbo. The corresponding answer is:

```
The first pictures of a planet outside of our own solar system were
taken by the Hubble Space Telescope in 2008. The planet, named
Fomalhaut b, orbits the star Fomalhaut, which is located about 25
light-years away from Earth in the constellation Piscis Austrinus.
```

We can see that the above answer is incorrect. We prompt the fact into the LLM, then the answer is:

```
The Very Large Telescope took the very first pictures of a planet
outside of our own solar system.
```

---

[2] https://www.theverge.com/2023/2/8/23590864/google-ai-chatbot-bard-mistake-error-exoplanet-demo

This indicates that prompting works. We further ask `gpt-3.5-turbo` to generate some sufficient statements for the fact. The following is its output:

```
1. The Very Large Telescope made history by capturing the first-ever
images of a planet beyond our solar system.
2. Thanks to the advanced technology of the Very Large Telescope,
scientists were able to capture the first-ever photographs of an
exoplanet.
3. The groundbreaking images captured by the Very Large Telescope
marked a major milestone in the study of exoplanets.
4. The discovery of a planet outside of our solar system was made
possible by the cutting-edge capabilities of the Very Large Telescope.
5. The Very Large Telescope's ability to capture images of distant
celestial bodies allowed scientists to observe an exoplanet for the
first time in history.
```

We also evaluate separately using the above statements as a prompt to `gpt-3.5-turbo`, and either one of them can lead to the correct answer. We provide detailed answers as follows.

```
1. The Very Large Telescope took the very first pictures of a planet
outside of our own solar system.
2. The Very Large Telescope took the very first pictures of a planet
outside of our own solar system.
3. The Very Large Telescope took the very first pictures of a planet
outside of our own solar system.
4. The Very Large Telescope took the very first pictures of a planet
outside of our own solar system.
5. The Very Large Telescope took the very first pictures of a planet
outside of our own solar system.
```

We can see, in this case, that either one of the statements generated by `gpt-3.5-turbo`, is a sufficient instance and a necessary instance to answer *What took the very first pictures of a planet outside of our own solar system?*

We further evaluate the performance of `gpt-j-6b`. Without any prompt (i.e., in the zero-shot setting), its answer is:

```
The Hubble Space Telescope.
```

We then prompt the above five statements into `gpt-j-6b`, then the answer is:

```
1. The first-ever images of a planet beyond our solar system have
been captured by the Very Large Telescope in Chile.
2. The Hubble Space Telescope.
3. A team of astronomers led by the University of Arizona's Michael
Liu.
4. The Hubble Space Telescope.
5. The Very Large Telescope.
```

We can see that only the first or the fifth statement is a sufficient instance. Combining the results of `gpt-j-6b` and the results of `gpt-3.5-turbo` can verify the core insight of our paper: *the measurement over a plug-in instance should consider what LLM is in use.*

Furthermore, we also evaluate the performance of three GPT variants used in the paper. We begin by evaluating the zero-shot performance of `gpt-neo-1.3B` (denoted as `NEO` in the experiment), and its result is:

```
The first pictures of a planet outside of our own solar system were
taken by the Voyager 1 spacecraft in 1977
```

We then prompt the above five statements into `gpt-neo-1.3B`, then the answer is:

```
1. The very large telescope was built in the early 1990s by the
European Southern Observatory (ESO) in Chile.
The Very Large Telescope.
2. The Very Large Telescope in Chile.
3. The Very Large Telescope (VLT) in Chile.
4. The Very Large Telescope.
5. The Very Large Telescope in Chile
```

The above results show that either one of the latter four statements is a sufficient instance. The results of `gpt2-large` (denoted as LAR in the experiment) show that neither of the five statements is a sufficient instance:

```
1. The very large telescope was built in the early 1990s by the
European Southern Observatory (ESO) in Chile.
The Very Large Telescope.
2. The Hubble Space Telescope.
3. The first pictures of a planet outside of our own solar system
were taken by the Hubble Space Telescope in 1990.
4. The Hubble Space Telescope.
5. The very first pictures of a planet outside of our own solar
system were taken by the Hubble Space Telescope.
```

The results of `gpt2-medium` (denoted as MED in the experiment) show that only the fourth statement is not a sufficient instance:

```
1. The Very Large Telescope.
2. The Very Large Telescope.
3. The Very Large Telescope.
4. The Hubble Space Telescope.
5. The Very Large Telescope.
```

All the above results verify that *quality* of one demonstration should be LLM-specific, which is the key idea of our paper.

