# OpenReview forum: "Parsimonious Demonstrations and Fine-Tuning for Large Language Models"
_ICLR.cc/2024/Conference — ICLR 2024 Conference Withdrawn Submission_

### Official Review · Reviewer_2FTd · 2023-10-31

**Soundness:** 3 good
**Presentation:** 3 good
**Contribution:** 2 fair
**Rating:** 5
**Confidence:** 4

**Summary:**

Quality of the result from LLMs is excessively dependent on the input prompt. Providing the right set of example as demonstrations helps boost the performance significantly.

But due to the limitation on the context length, one needs to limit the number of examples that can be provided to the model as demonstrations. The paper presents an algorithm to select the right set of examples as demonstrations to the model that can achieve the better than some of the commonly used methodologies.

The effectiveness of the methodology is supported by experiments that show that just by selecting the right set of examples, the model can significantly improve performance on various tasks.

The paper additionally proposes to extend this to finetuning. Essentially, train the model on only a subset of high quality examples instead of all train examples. Results in the finetuning setup are also encouraging.

**Strengths:**

- The problem being tackled is pretty important given the increasing size of LLMs makes finetuning on downstream tasks pretty difficult.
- The paper standardizes brute force techniques used to select the right set of examples as demonstrations automatic some key decisions that need to be taken in such selections.
- The results are pretty encouraging on k-shot as well as finetuning setup.

**Weaknesses:**

- The method is pretty complicated, and pretty expensive if the train dataset is large.
- Selecting every pair for the necessity check makes it pretty expensive.
- It might be better to use adaptation techniques like LORA and prompt tuning, which might have lower cost than optimizing for the right set of demonstrations, even though the former has training involved. But I agree that the model is not always accessible for finetuning.
- The method proposed is expensive to a point that even though the results are good, the method might rarely be applied in practical use-cases.

**Questions:**

- Was there any comparison done on compute cost of selecting the right demonstrations vs adaptation techniques?
- Comparison on quality attainable using soft prompt tuning or similar adaptation methods vs the proposed method would also be good.

---

### Official Review · Reviewer_jRpK · 2023-11-01

**Soundness:** 2 fair
**Presentation:** 3 good
**Contribution:** 2 fair
**Rating:** 5
**Confidence:** 3

**Summary:**

This paper proposes FEEDER, a data miner that selects a parsimonious set of demonstrations for large language models (LLMs). FEEDER evaluates the sufficiency and necessity of each demonstration based on the LLMs used, and uses tree-based search algorithms to identify the parsimonious set. The paper shows that the parsimonious set can improve the performance and efficiency of LLMs in both in-context learning and fine-tuning settings.

**Strengths:**

1. The paper introduces a novel data miner, FEEDER, that selects a parsimonious set of demonstrations for large language models (LLMs).
2. The paper shows that the parsimonious set can improve the performance and efficiency of LLMs in both in-context learning and fine-tuning settings.
3. The paper devises tree-based search algorithms to identify the parsimonious set efficiently and provides theoretical analysis and proofs.

**Weaknesses:**

1. My main concern is that the paper does not compare FEEDER with other core-set selection methods or demonstrate its generalization to other tasks and domains. Actually, there have been many works for both in-context learning and fine-tuning that utilize retrieval from a large set.
2. The paper assumes that sufficiency follows a transitive relationship among sets, which may not always hold in practice.
3. The paper does not address the scalability and robustness issues of FEEDER when dealing with large and noisy datasets.

**Questions:**

How do you compare FEEDER with other core-set selection methods in terms of computational complexity and scalability? What are the advantages and limitations of your tree-based search algorithms?

---

### Official Review · Reviewer_HTu1 · 2023-11-01

**Soundness:** 2 fair
**Presentation:** 3 good
**Contribution:** 2 fair
**Rating:** 3
**Confidence:** 3

**Summary:**

This paper introduces FEEDER, a tree-search algorithm designed to optimally select demonstrations for LLMs. Unlike other algorithms that overlook the specifics of the LLM being used, FEEDER evaluates demonstrations for their "sufficiency" and "necessity" with respect to the particular LLM in question. Consequently, it extracts a concise set of highly informative samples from a given training dataset. Designed for both in-context learning and fine-tuning scenarios, empirical tests across multiple datasets and LLMs confirm FEEDER's efficacy as a pre-filter, reducing the size of the training data while either preserving or enhancing the model's performance.

**Strengths:**

1. FEEDER's approach of tailoring demonstration selection based on the specific LLM in use is both innovative and logical.
2. The empirical results include a variety of datasets and LLMs, giving a comprehensive overview of FEEDER's effectiveness.
3. The provision to incrementally update FEEDER in response to growing datasets enhances its practical utility.

**Weaknesses:**

1. Despite empirical evidence supporting FEEDER's value as a pre-filtering algorithm, the paper does not provide a direct comparison with existing state-of-the-art demonstration selection methodologies. As indicated in Table 1, when FEEDER is used solely (i.e., comparing D_FEED + RAN to D_TRAIN + SIM or D_TRAIN + DIV), it often underperforms relative to other methods. Furthermore, FEEDER's marginal or even negative enhancements as a pre-filter to SIM and DIV, coupled with its computational demands, might render SIM and DIV more favorable options due to their efficiency and simplicity.
2. The research primarily focuses on text classification datasets. Given that one of the LLM's standout features is its reasoning and generation capabilities, it remains uncertain how well FEEDER would adapt to tasks or domains beyond text classification.

**Questions:**

1. How does the size of the parsimonious set generated by FEEDER compare to the original training data volume? Given that demonstrations are typically restricted to a specific count (e.g., 5 or 10), is there scope to refine the algorithm to pinpoint, for instance, the top-5 or top-10 most informative samples?
2. Since the search algorithm aims to optimize training data performance, could this potentially lead to overfitting? Would integrating a validation set serve as a solution against such overfitting?

---

### Official Review · Reviewer_5TLU · 2023-11-10

**Soundness:** 3 good
**Presentation:** 3 good
**Contribution:** 2 fair
**Rating:** 5
**Confidence:** 3

**Summary:**

This paper proposes a framework ‘FEEDER’ that evaluates “sufficiency” and “necessity” to select demonstrations effectively. To find sets that are both sufficient and necessary, called parsimonious sets, without evaluating all possible subsets the authors devise a tree-based algorithm. An experiment is conducted on in-context learning and fine-tuning tasks and shows competitive results.

**Strengths:**

- The authors addressed the important question and explained their approach thoroughly.
- Empirical results show that the use of parsimonious sets contributes to improved performance.
- FEEDER reduces computational costs in practical settings, such as incremental updates, avoiding the need for recomputing all sets.

**Weaknesses:**

There are two major concerns:

- **Computation Cost:** While attempts to reduce computational costs have been made, the expense remains a significant consideration. Additionally, a comparison of computational cost measurements with other approaches is lacking.
- **Experiment Setting:** The choice of benchmark methods, such as random, similarity, and diversity, is questionable. Advanced methods like incorporating diversity and similarity or considering recent works with similar motivations could enhance the quality of the experiment. Also, the experiment is only conducted on text classification, it can be extended to diverse tasks like QA and show applicability.

**Questions:**

The authors argue that "previously used metrics should be thoroughly revised in the new era of LLMs because they measure each data instance regardless of the LLMs in use." However, some works use frozen LLMs for measurement, similar to your approach. Could you clarify the exact meaning of 'LLMs in use'?